# Gone With the Bits: Revealing Racial Bias in Low-Rate Neural Compression for Facial Images

## Abstract

Neural compression methods are gaining popularity due to their impressive rate-distortion performance and their ability to compress data to extremely small bitrates, below 0.1 bits per pixel (bpp). As deep learning architectures, these models are prone to bias during the training process, potentially leading to unfair outcomes for individuals in different groups. In this paper, we present a general, structured, scalable framework for evaluating bias in neural image compression models. Using this framework, we investigate racial bias in neural compression algorithms by analyzing 7 popular models and their variants. Through this investigation we first demonstrate that traditional distortion metrics are ineffective in capturing bias in neural compression models. Next, we highlight that racial bias is present in all neural compression models and can be captured by examining facial phenotype degradation in image reconstructions. Additionally, we reveal a task-dependent correlation between bias and model architecture. We then examine the relationship between bias and realism in the image reconstructions and demonstrate a trade-off across models. Finally, we show that utilizing a racially balanced training set can reduce bias but is not a sufficient bias mitigation strategy.

## 1 Introduction

Lossy image compression aims to accurately represent images using a minimal number of bits while maintaining their perceptual quality in reconstructions. This area has been the focus of extensive research for the past 40 years, and image encoders/decoders ("codecs") such as JPEG (Wallace, 1991), BPG (Bellard, 2014), and even the latest hand-engineered codec in VVC (Bross et al., 2021) have been crucial enabling technologies in the modern digital world. Despite the widespread adoption in everyday use, traditional codecs are insufficient for extreme scenarios with low-bandwidth availability, such as space (Gao et al., 2023), underwater (Li et al., 2023), low-power communication systems Ez-Zazi et al. (2018) and low-latency systems Hu & Chen (2021). These extreme scenarios impose a very narrow information bottleneck that limits the reconstruction quality of traditional codecs. In recent years, neural network-based compression ("neural compression") has emerged as a popular compression method that enables image compression under extremely low-bitrate scenarios. Early works in this field (Toderici et al., 2015; 2017) utilize recurrent neural networks, while many subsequent studies have employed VAE-based architectures (Ballé et al., 2018; Townsend et al., 2019; Duan et al., 2023a;b). Recent studies explore leveraging modern generative architectures such as GANs (Agustsson et al., 2019; Mentzer et al., 2020) and Diffusion (Yang & Mandt, 2023) to promote higher levels of realism in reconstructions.

The goal of this paper is to examine potential unwanted biases in low-rate neural compression models. We consider a scenario where we train a neural compression model, specialized for human faces, to attain a very low bitrate. Regardless of the compression method used, image reconstructions at low bitrates will inherently suffer from significant distortion due to the insufficient number of bits used to represent images. The central question we pose is the following: *when we train a neural network models to compress human faces with low bitrates, would the model degrade facial images equally across different demographic groups? Or, would it prioritize accurately reconstructing one racial group's faces, at the expense of sacrificing image qualities of another racial group when the information bandwidth is limited?* Such biased and unfair performance of neural compression can

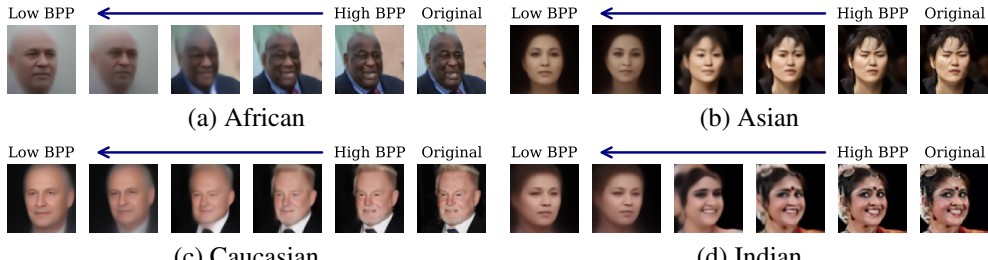

Figure 1: All the neural compression models in our evaluation exhibit **bias in skin type** for **African racial group**. Examples are from the *QRes* Duan et al. (2023b) model. As compression bitrate reduces, African faces gradually experience skin-lightening effects, while other racial groups are impacted less. Our novel evaluation approach with phenotype classifiers quantifies how different phenotypes degrade and highlights bias in this process.

have a significant impact on people of marginalized groups, especially in extreme and high-risk scenarios where low-rate compression schemes are deployed (e.g., delaying rescue operations due to inaccurate facial images transmitted in a warzone).

This question is inspired by a line of research that studies related questions. In (Yucer et al., 2022a), the authors investigate bias in face image compression using the traditional JPEG scheme and show unequal performance in facial recognition tasks across different racial groups. Recent works (Jalal et al., 2021; Laszkiewicz et al., 2024; Tanjim et al., 2022) also looked at biases of image construction using neural networks. Although these works differ from our setting in that they start with downsampled or heavily corrupted facial images and use neural networks only for denoising or super-resolution, we see a fundamental connection to our work: downsampling or adding noise can be viewed as imposing a narrow information bottleneck, similar to compression. In these settings, it was shown that the reconstructed images often show a specific type of distortion—African American faces are frequently reconstructed to appear more Caucasian, while Caucasian faces largely retain their original features—a phenomenon referred to as the "White Obama" problem (Jalal et al., 2021; Laszkiewicz et al., 2024). Despite these works, to the best of knowledge, *our work is the first to examine bias in neural compression models, consisting of a neural network encoder and decoder.*

To comprehensively explore our central question, we propose the following research questions: **RQ1**. Do neural compression models exhibit bias, and how can we quantify this bias? **RQ2**. How does bias vary across different model architectures? **RQ3**. Does using a balanced dataset reduce or eliminate bias? To answer the research questions, we design a general framework and metric to evaluate bias in neural image compression models and perform a detailed analysis of racial bias in facial reconstructions using state-of-the-art models. We also investigate how different model architectures impact bias and assess the influence of training data distribution by using racially balanced datasets, leading to the following key observations:

- Traditional image distortion cannot effectively capture neural compression bias, while our proposed framework using classifiers, is able to highlight significant *skin type* bias for images in the African racial group, supporting visual observation of image reconstructions.

- We reveal a phenotype-dependent correlation between bias and model architectures. Specifically, diffusion-based models exhibit severe *skin type* bias for the African group, while the GAN-based model does not.

- Leveraging a racially balanced training dataset can reduce bias in certain cases but not in others, motivating further exploration into the development of balanced datasets and algorithmic bias mitigation methods.

## 2 RELATED WORK

**Fairness in Image Compression**  Our work is closely related to Yucer et al. (2022a), which studies the impact of JPEG compression on facial verification and identification tasks and the amount of adverse impact of JPEG compression on different racial and phenotype-based subgroups. They define bias as the different amount of downstream task performance degradation across groups. They find

phenotype groups of darker skin tones, wide noses, curly hair, and monolid eye shapes suffer the most adverse impact in the facial recognition tasks. Hofer & Böhme (2024) study neural compression model reconstructions through visual inspection and gives a taxonomy of "mis-compressions", which they define as errors in semantic information after neural compression. Our work not only studies bias in neural compression through visual inspection but also aims to capture bias in a structured and scalable approach through a facial phenotype classifier. We see this as a first step towards systematically evaluating and mitigating bias in neural image compression models.

**Fairness in Image Denoising and Upsampling**   Stemming from the "White Obama" problem, fairness has been explored across image upsampling, denoising, and superresolution models. Menon et al. (2020), the authors of the original model which suffers from the "White Obama" problem, conduct an investigation concluding the bias is likely induced during the creation of the StyleGAN which they adopt for their task. Jalal et al. (2021) design novel definitions of fairness for image upsampling tasks and highlight fairness-accuracy tradeoffs for these types of models. Tanjim et al. (2022) examine the disappearance of minority attributes such as eye-glasses and baldness during image-to-image generation. They also propose a contrastive learning framework to improve upon bias in existing image-to-image translation models. Laszkiewicz et al. (2024) aim to study and benchmark the fairness in face image upsampling, demonstrating bias when imbalanced datasets are used while training these upsampling methods.

**Fairness in Face Analysis**   The processing of facial images is utilized across various domains, including face recognition, facial biometrics, and facial expression recognition. Fairness in such systems is crucial and has been studied in various aspects of the face and biometric analysis (Drozdowski et al., 2020; Vangara et al., 2019; Serna et al., 2019). Buolamwini & Gebru (2018) evaluated commercial gender classification tools and identified that darker-skinned females suffer from significantly higher misclassification rates than lighter-skinned males. Klare et al. (2012) found that various face recognition systems exhibited the poorest performance on cohorts comprising females, Black individuals, and those aged 18-30. Motivated by the imbalanced distribution of datasets used for facial expression detection, Xu et al. (2020) investigate biases across gender, race, and age groups, and propose methods to mitigate these biases in such models.

## 3   PROBLEM DEFINITION AND METHODS

Overall, our goal is to develop a framework to evaluate and quantify bias in neural compression image reconstructions. In Section 3.1 we provide an overview of neural image compression. In Section 3.2 we define a general bias metric to evaluate bias in neural compression reconstructions. In Section 3.3 we highlight a specific instance of the bias metric, using a phenotype classifier to examine bias.

### 3.1   NEURAL IMAGE COMPRESSION

Neural compression models consist of an encoder $g_{enc} : \mathcal{X} \to \mathcal{Z}$ and a decoder $g_{dec} : \mathcal{Z} \to \mathcal{X}$, each built from learnable network layers. For each input image $x \in \mathcal{X}$, the encoder is used to obtain the latent space output $z$, which is then quantized to $\hat{z}$ and compressed losslessly to a bitstream. This bitsream is then decompressed to $\hat{z}$ and passed through the decoder to provide the decoded image $\hat{x}$. Overall, the goal for neural compression models is to minimize

$$\mathcal{D}(x, \hat{x}) + \lambda \mathcal{R}(\hat{z}) \tag{1}$$

where $\mathcal{D}(x, \hat{x})$ is the distortion, $\mathcal{R}(\hat{z})$ is the compression bitrate, and $\lambda$ acts as the Lagrange multiplier that balances the rate-distortion trade-off. Distortion is typically measured using the mean squared error between the original image and the reconstruction while the bitrate is bounded using the entropy of the quantized latent $\hat{z}$.

### 3.2   EVALUATING BIAS IN NEURAL COMPRESSION

We aim to define a scalable, general framework to analyze the bias in neural compression models. Let $\mathcal{D} = \{(x_i, y_i, a_i)\}_{i=1}^n$ be our dataset, where $x_i \in \mathcal{X}$ is our image, $y_i \in \mathcal{Y}$ is a label corresponding to a physical attribute of the image, and $a_i \in \mathcal{A}$ is a protected attribute. Our goal is to examine how the quality of reconstructions of $x_i$ differ across $\mathcal{A}$. First, given a pretrained encoder and decoder, we can obtain the reconstructed dataset $\widehat{\mathcal{D}}(g_{enc}, g_{dec}) = \{(\hat{x}_i, y_i, a_i)\}_{i=1}^n$ and consider

a general loss metric $\mathcal{L}(\mathcal{D}, \widehat{\mathcal{D}}(g_{\text{enc}}, g_{\text{dec}}))$ which is designed to evaluate the quality of the reconstruction (e.g. distortion metric, downstream task performance). We include the original dataset $\mathcal{D}$ in the general loss metric as some metrics (e.g distortion) compare reconstructions to original images. Note that this original dataset is not needed in all loss metrics and we omit it when it is not used. Now, from this general loss metric, we can derive a conditional loss metric

$$\mathcal{L}(\mathcal{D}, \widehat{\mathcal{D}}(g_{\text{enc}}, g_{\text{dec}})|a) = \mathcal{L}(\mathcal{D}, \widehat{\mathcal{D}}_a(g_{\text{enc}}, g_{\text{dec}})) \tag{2}$$

where $\widehat{\mathcal{D}}_a(g_{\text{enc}}, g_{\text{dec}}) = \{(\hat{x}_i, a_i, y_i) \in \widehat{\mathcal{D}}(g_{\text{enc}}, g_{\text{dec}})|a_i = a\}$.

Using this conditional loss, we can define bias to be

$$\text{Bias} \triangleq \max_{a,b \in A}[\mathcal{L}(\widehat{\mathcal{D}}(g_{\text{enc}}, g_{\text{dec}})|a) - \mathcal{L}(\widehat{\mathcal{D}}(g_{\text{enc}}, g_{\text{dec}})|b)]. \tag{3}$$

This bias term represents the maximum difference in loss across groups in $\mathcal{A}$. Surprisingly, different selections of the loss function yield different insights into the bias of the neural compression architectures. As we will show in the following sections, traditional distortion metrics show no apparent bias, while the accuracy of a phenotype classifier highlights significant bias across different racial groups (Section 4.2).

### 3.3 Bias Evaluation with a Phenotype Classifier

From visual inspection of image reconstructions, we identify key facial phenotypes (e.g., skin color, eye shape) can get degraded under low-rate neural compression. To systematically quantify phenotype degradation induced by the neural compression architecture, accurate labels are required for image reconstructions. Hand-labeling the phenotypes in the reconstructed images would be the most accurate way to obtain these labels, but it is not a scalable procedure for large image datasets. Therefore we propose to use a neural-network-based phenotype classifier as a proxy of human evaluation. Additionally, using a classifier to identify biases across different racial groups offers valuable insights into the potential disparities that may emerge when reconstructed images are used in subsequent deep-learning tasks. Previous studies (Jalal et al., 2021; Tanjim et al., 2022; Laszkiewicz et al., 2024) have investigated the use of phenotype classifiers to assess or mitigate bias in facial images. These existing metrics, however, consider super-resolution-specific problem settings and do not necessarily transfer to the image compression domain, as we highlight in Example 3.1.

First, given a dataset $\mathcal{D}$ where $\mathcal{A}$ is the set of racial groups (e.g {African, Asian, Caucasian, Indian}), and $\mathcal{Y}$ is the set of possible phenotype labels (e.g {bald, curly hair, straight hair, wavy hair} for *hair type*), we split into $\mathcal{D}_{\text{train}}$ and $\mathcal{D}_{\text{test}}$ and use $\mathcal{D}_{\text{train}}$ to train a classifier $f : \mathcal{X} \to \mathcal{Y}$ to predict the phenotype labels (this can be a binary or multiclass classification task). Then, given a pretrained encoder and decoder at bitrate $r$, the original test dataset $\mathcal{D}_{\text{test}}$ is compressed to the bitrate $r$ and reconstructed to $\widehat{\mathcal{D}}_{\text{test}}^r(g_{\text{enc}}, g_{\text{dec}}) = \{(\hat{x}_i, y_i, a_i)\}_{i=1}^n$. To measure phenotype degradation at the given rate, we define our loss function to be the error rate of $f$ on $\widehat{\mathcal{D}}_{\text{test}}^r$:

$$\text{Err}(\widehat{\mathcal{D}}_{\text{test}}^r(g_{\text{enc}}, g_{\text{dec}})) = \mathbb{P}_{(\hat{x},y)\sim\widehat{\mathcal{D}}_{\text{test}}^r(g_{\text{enc}}, g_{\text{dec}})}(f(\hat{x}) \neq y). \tag{4}$$

The conditional loss then becomes:

$$\text{Err}(\widehat{\mathcal{D}}_{\text{test}}^r(g_{\text{enc}}, g_{\text{dec}})|a) = \mathbb{P}_{(\hat{x},y,a)\sim\widehat{\mathcal{D}}_{\text{test}}^r(g_{\text{enc}}, g_{\text{dec}})}(f(\hat{x}) \neq y|A = a). \tag{5}$$

By defining the loss function to be the error rate of the phenotype classifier, our bias metric directly becomes *accuracy disparity*, the maximum difference of accuracy across all groups (due to the standard relationship between error rate and accuracy). Given a rate $r$, an encoder $g_{\text{enc}}$, and a decoder $g_{\text{dec}}$, the bias metric is defined as:

$$\text{Bias}(\widehat{\mathcal{D}}_{\text{test}}^r(g_{\text{enc}}, g_{\text{dec}})) \triangleq \max_{a,b \in \mathcal{A}}[\text{Acc}(\widehat{\mathcal{D}}_{\text{test}}^r(g_{\text{enc}}, g_{\text{dec}})|a) - \text{Acc}(\widehat{\mathcal{D}}_{\text{test}}^r(g_{\text{enc}}, g_{\text{dec}})|b)] \tag{6}$$

where $\text{Acc}(\widehat{\mathcal{D}}_{\text{test}}^r(g_{\text{enc}}, g_{\text{dec}})|a) = 1 - \text{Err}(\widehat{\mathcal{D}}_{\text{test}}^r(g_{\text{enc}}, g_{\text{dec}})|a)$. This definition of bias is derived from a popular fairness metric, *accuracy parity*, in which equal accuracies across all groups imply fairness in a classifier (Berk et al., 2017; Zafar et al., 2017). The motivation behind the selection of this bias definition can be observed in the following example.

**Example 3.1** *Let $\mathcal{A}$ be the set of races {African, Caucasian} and let $\mathcal{Y} = \{light\ skin,\ dark\ skin\}$. In this case, the conditional error in Equation 5 captures the error rate of the skin color classification in the reconstructed image space for each group. When these conditional error rates are similar across $\mathcal{A}$, the skin colors switch equally for both groups in $\mathcal{A}$. When these values are different across $\mathcal{A}$, one race suffers from a skin color switch significantly more than another. Thus, the bias metric presented in Equation 6 captures a more descriptive insight into what leads to race flipping than traditional metrics, which may only capture the frequency of the race flipping (Jalal et al., 2021). By changing $\mathcal{Y} = \{monolid\ eyes,\ non-monolid\ eyes\}$ or any other phenotype, we can gain additional insight into how specific phenotypes get lost at different rates across each group in $\mathcal{A}$.*

We acknowledge that these phenotype classifiers can be biased themselves. Using our framework, we can compute the accuracy disparity of our phenotype classifier on the original raw images. We present these "raw accuracies" in Section 4.2 to provide context into the bias induced by the compression model. Additionally, we address the potential distribution shifts induced by the neural compression models in Section 4.2.

## 4 EXPERIMENTS AND EVALUATION

### 4.1 EXPERIMENTAL SETUP

**Neural Compression Models**    In this paper, we evaluate a diverse collection of neural image compression models across different bitrates. An overview of our models is shown in Table C.1. We evaluate three fixed-rate models, *Hyperprior* Ballé et al. (2018), *Joint* Minnen et al. (2018), and *GaussianMix-Attn* Cheng et al. (2020). All of these models are trained towards a fixed trade-off between rate and distortion as highlighted in Equation 1. We train these models to five operational bitrates. The model proposed in the *QRes* paper Duan et al. (2023b) is a progressive decoding model that supports encoding images to 12 bitrates with one trained model. This is achieved by encoding only a subset from all the available latent variables. We follow this approach and encode images to 5 different bitrates with progressive decoding. The *VarQRes* model Duan et al. (2023a) is a variable rate compression model. The network is trained to operate in a range of rate-distortion trade-off points. Additionally, we consider two models which leverage attributes of popular generative models. The *HiFiC* model Mentzer et al. (2020) combines GANs with neural compression by introducing a discriminator conditioned on the latent variable following the decoder. The *CDC* model (Yang & Mandt, 2023) is a conditional diffusion model which closely resembles a diffusion-based autoencoder. In addition to the standard *CDC* model, we consider two variants, *CDC-L2* in which an auxiliary loss term is added that directly captures the distortion between the original image and the generated image, and *CDC-LPIPS*, where the model adds an optional realism loss measured by LPIPS (Zhang et al., 2018). We describe model implementations and training details in Appendix C.2.

**Phenotype Classifier**    To study phenotype degradation in decoded images from neural compression, we use the Racial Faces in the Wild (RFW) dataset (Wang et al., 2019) and a recently released facial phenotype annotation dataset specifically for RFW (Yucer et al., 2022b). This annotation dataset provides labels for six phenotype categories—skin type, eye type, hair type, hair color, lip type, and nose type—across four racial groups: African, Indian, Asian, and Caucasian. Skin types are labeled into 6 classes according to Fitzpatrick Skin Types (Fitzpatrick, 1988). Eye types are labeled as monolid or non-monolid. Nose types are labeled wide or narrow depending on nasal breadth. Hair types are labeled into 4 groups: bald, curly, straight, and wavy. Lip types are labeled as either full or small. Hair colors are labeled red, grey, black, blonde, and brown. The distribution of phenotypes across these racial groups is depicted in Figure A.1.

We train individual classifiers for each phenotype classification task (e.g. one model for eye type classification, one for hair type classification, etc.), leading to either a binary or multi-class classification task. Training details for the phenotype classifiers can be found in Appendix C.1. When measuring bias, we utilize the racial groups as our sensitive attribute, defining $\mathcal{A}$ as the set of all racial groups. When performing inference for multi-class classification tasks hair color and hair type, we group the three most dominant classes for each group. For skin type, we group all classes that make up at least 5% of the group. This allows us to evaluate the extent to which phenotypes flipped to those not prevalent in the racial group of the raw image.

**Datasets**    We train all neural compression models on the CelebA (Liu et al., 2018), FaceARG (Darabant et al., 2021), and FairFace (Kärkkäinen & Joo, 2019) datasets. These datasets are chosen

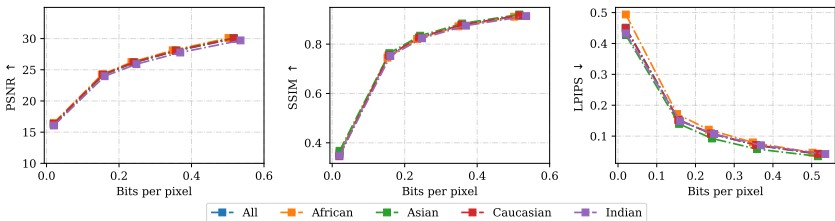

Figure 2: Traditional rate-distortion metrics (PSNR, SSIM, and LPIPS) for the *Joint* model trained on the CelebA dataset, shown for each race and the overall dataset. The rate-distortion curves are nearly identical across all races for PSNR and SSIM, which contrasts with the findings from the qualitative analysis. While the LPIPS curve for the African group is slightly higher than for other races, it fails to fully reflect the disparities observed in the qualitative analysis.

to make comparisons between the impact of racially balanced and imbalanced training sets. The CelebA dataset has a significantly imbalanced racial composition with more than 70% of the images from the white racial group (Kärkkäinen & Joo, 2019). Additionally, we leverage the FaceARG dataset and the FairFace Dataset to investigate the effect of a balanced training dataset. FaceARG is a large-scale dataset containing over 175,000 facial images, each labeled with age, gender, race, and ethnicity. The dataset features a relatively balanced distribution of images across four different racial groups: African, Asian, Caucasian, and Indian. The FairFace dataset contains over 100,000 images with a balanced racial composition across seven race groups: White, Black, Indian, East Asian, Southeast Asian, Middle Eastern, and Latino. All images are down-sampled to 64x64 resolution. Finally, to quantify the relationship between realism and bias, we utilize the DemogPairs dataset (Hupont & Fernández, 2019) as a reference to compute FID scores of the decoded images.

### 4.2 Do neural compression models exhibit bias? How can we quantify it?

Our initial observation of the skin type phenotype being lost in darker-skinned individuals, as illustrated in Figure 1, prompts us to investigate the potential biases present in various neural compression models across different compression rates. We aim to quantify the potential biases associated with preserving different phenotypes across different races in images compressed using various neural compression models.

**Traditional Distortion Metrics**  To quantify the aforementioned bias, we first investigate how traditional distortion metrics reflect potential bias in neural compression models. We conduct two experiments using PSNR, SSIM and LPIPS as the loss functions in Equation 3 and present the results for the *Joint* model trained on the CelebA dataset in Figure 2. The traditional distortion metrics results for other studied models are presented in Appendix B. The rate-distortion curves highlight that distortion values across each race are nearly identical to that of the overall dataset, suggesting that facial images in different race groups are distorted by similar amounts at similar rates. The LPIPS curve for African faces sits slightly higher than the others but does not capture the extent of change seen in the qualitative analysis. This indicates that traditional distortion metrics are not suitable for capturing the bias in these neural compression architectures, which motivates the need for an alternative metric to capture this bias more effectively.

**Phenotype Classification Metric**  To more accurately quantify potential biases in the compressed images, we employ the bias metric defined in Equation 6 and present the classification results for the *skin type* phenotype in the *Joint* model trained on the CelebA dataset, as shown in Figure 3(a). The figure reveals a significant decline in classification accuracy for individuals in the African group at low bitrates, while accuracy for images from other racial groups remains relatively stable. This disproportionate drop in accuracy for the African group leads to an increased level of bias as the bitrate decreases, aligning with our qualitative analysis. These findings indicate that using Equation 6 to quantify bias values provides a more precise assessment of the biases in compression.

To further explore how bias is amplified at varying compression rates, we plot the bias values across different phenotypes for the *Joint* model in Figure 3(b). We observe that the bias in the classification of *skin type*, *eye type*, and *hair type* increases as compression rates decrease, while other phenotypes display relatively low bias throughout. Specifically, the rise in bias for *skin type* and *eye type* is primarily driven by a disproportionate drop in accuracy for the African group, while the increased

Table 1: **Phenotype Classification on Raw Data.** Phenotype classification accuracies and bias (Equation 6) values on raw data rounded to 2 decimal places. Largest values for each task **bolded**, smallest values *italicized*.

| Race | Skin Type | Eye Type | Nose Type | Lip Type | Hair Type | Hair Color |
|------|-----------|----------|-----------|----------|-----------|------------|
| Asian | 0.92 | *0.78* | 0.59 | 0.76 | **0.99** | 0.90 |
| African | *0.89* | **0.98** | **0.83** | 0.71 | *0.85* | **0.96** |
| Caucasian | **0.96** | 0.93 | *0.57* | **0.83** | 0.96 | *0.77* |
| Indian | 0.92 | 0.96 | 0.65 | *0.57* | 0.96 | 0.86 |
| Bias | 0.07 | 0.20 | 0.26 | 0.27 | 0.14 | 0.20 |

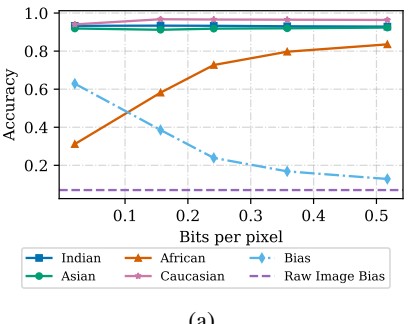

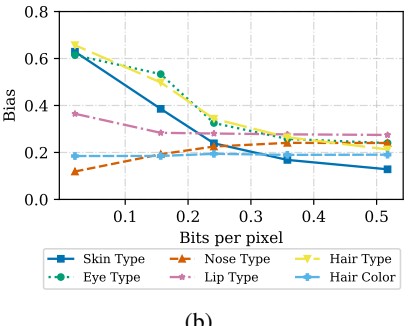

(a)                              (b)

Figure 3: (a) Bias for *Skin Type* across different races for *Joint* reconstructions trained on the CelebA dataset. (b) As the bitrate is lowered, bias increases for *Skin Type*, *Eye Type*, and *Hair Type*, while remaining relatively level for other phenotypes.

bias for *eye type* is linked to a decline in accuracy for the Asian group. This bias trend is consistently observed, to varying degrees, across all other neural compression architectures studied. A more in-depth analysis of these differences in bias trends is provided in Section 4.3.

**Evaluation of Phenotype Classifiers** Following the quantification of bias from the phenotype classification framework, we evaluate the performance of our phenotype classifier to validate its ability to accurately capture the target phenotypes in the raw images as well as under distribution shifts caused by neural compression models. As outlined in Section 4.1, we train separate classifiers for each phenotype using raw RFW image data, and use these classifiers to assess phenotype preservation across various compression rates. We report the classifiers' accuracies for the specific classification tasks on the raw RFW images in Table 1. We observe that classifiers trained on raw images exhibit varying initial biases for different tasks; however, the changes in bias values across different compression rates do not adhere to a consistent pattern. For example, as previously noted, the increasing bias trend linked to the disproportionate decline in accuracy for the *skin type* classification in African images is evident across all the neural compression models studied (Appendix D). In contrast, the initial bias for *hair color*, which begins at a higher value, remains relatively stable across various compression rates and models. This suggests that the classifiers effectively capture the desired phenotypes, indicating that the observed bias cannot be solely attributed to the initial bias of the model. Moreover, using classifiers to capture biases in neural compression is likely to reflect the trends observed in machine learning models trained for downstream tasks on the compressed images, providing us with valuable insights early in the process.

Furthermore, to ensure the classifiers rely on relevant image features rather than spurious correlations, we analyze gradient-based saliency maps. Specifically, we generate smoothed saliency maps using SmoothGrad (Smilkov et al., 2017) for all classifications in our study. Figure 4 (a) displays the saliency maps produced by the *eye type* classifier on compressed images from the *VarQRes* model trained on the *CelebA* dataset. Highlighted regions indicate the areas of the image most influential in the final classification decision. The *eye type* classifier correctly identifies the important regions for determining the *eye type* phenotype. Additionally, to demonstrate the *skin type* classifier's sensitivity to changes in skin tone we present classification results on the compressed images from the

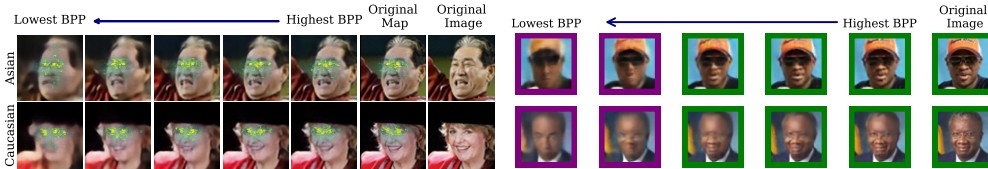

(a) *Eye type* saliency Maps for *VarQRes*  (b) African *skin type* classification

Figure 4: (a) Saliency maps for *eye type* classification at varying compression rates for Asian and Caucasian using the *VarQRes* model. Highlighted regions represent areas that had the most significant influence on the model's decision, showing emphasis on the eye area for classification. (b) *Skin type* classifier accurately captures shifts in skin color observed in African racial groups. Green borders indicate the correct classifier predicting the skin type to the ground truth label. Purple borders indicate the predicted skin type is lighter than the ground truth associated with the raw image.

*VarQRes* model trained on CelebA dataset in Figure 4 (b). We observe that as facial phenotypes get whitewashed at lower compression rates, the classifier accurately detects this shift and categorizes the skin tone accordingly.

We further explore the *skin type*, and *hair type* saliency maps to confirm the effectiveness of our classifiers in Appendix E.1. The *skin type* classifier effectively identifies the relevant area for classifying *skin type* in both Caucasian and African examples, focusing solely on the general facial region. However, the *hair type* classifier struggles to accurately locate the hair region in images of African individuals, while it successfully identifies the hair in Caucasian examples. We attribute this disparity primarily to the distribution of images and labels available for the African group. Our qualitative analysis present in Figure E.2 reveals that most of the randomly samples images of African individuals feature males with short hair or wearing headwear. This characteristic makes *hair type* classification more challenging for this racial group in contrast to other groups, such as Caucasians, where such limitations are less prevalent.

## 4.3 HOW DOES BIAS VARY ACROSS MODEL ARCHITECTURE?

We observe significant bias across neural compression models, which prompts us to investigate how bias arises differently across different neural compression models. To investigate this, we highlight the bias for different models for the *skin type* and *eye type* classification task in Figure 5. First, we observe that in the *skin type* classification task, there is a clear relationship between the model architecture and the bias we observe. The diffusion models (*CDC*, *CDC-L2* and *CDC-LPIPS*) appear to suffer from the most significant bias for the *skin type* classification task, followed by the VAE-based models (*Hyperprior*, *Joint*, *GuassianMix-Attn*, *QRes*, and *VarQRes*), and then the GAN-based model (*HiFiC*). This data supports the visual observations we make from the reconstructed images from the *HiFiC* model presented in Figure E.3, which provides further evidence of the phenotype classifier accurately capturing the desired phenotype. This architecture dependence trend reverses when we explore *eye type* classification. Here, the diffusion-based models experience the lowest amplification of bias while the GAN-based model experiences the highest level of bias. Again, the VAE-based models remain in the between the two types of generative models. These results suggest that the bias that different architectures vary across different classification tasks. We believe that future work can explore which specific properties of these model lead to specific types of bias and examine how leveraging properties from these architectures can help mitigate bias.

**Bias-Realism Relationship** In addition to directly comparing the bias, we systematically assess the relationship between bias and realism across neural compression models. This helps us understand whether models trade off these values and identify which objective each model can optimize. We quantify realism using Frechet Inception Distance (FID) (Heusel et al., 2017), while bias values are derived from Equation 6. FID provides statistical insight into how similar generated data is to a reference distribution. The reference distribution for FID is a set of real images to help capture how "realistic" the decoded images are. To ensure we are measuring realism with respect to general facial datasets, we utilize the Demogpairs dataset (Hupont & Fernández, 2019) as a reference for computing FID. This enables us to capture the fidelity of the reconstructions without spurious correlations to any of the datasets used during training.

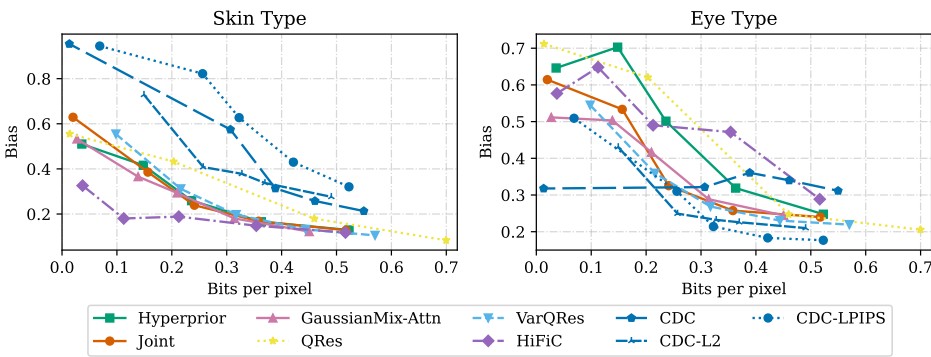

Figure 5: Bias in *Skin Type* and *Eye Type* across all neural compression models.

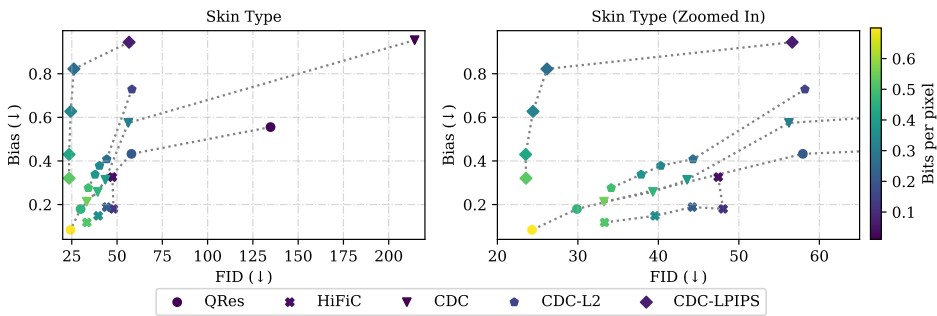

Figure 6: At high bitrates ($> 0.1$ bpp), bias and realism are correlated across all the models. At low bitrates ($< 0.1$ bpp), the trend is more sporadic.

We highlight that at lower FID values, there appears to be a positive correlation between bias and realism (Figure 6). As the realism deteriorates (FID increases), bias increases. These points mainly come from the intermediate bitrate regime. In the low bit rate regime, this trend degrades. Here, the relationship between bias and realism becomes much more sporadic. At lower levels of FID (higher realism), we can more clearly explore the relationship different neural compression models. We observe that *CDC-LPIPS* is able to preserve realism well as the bitrate is reduced while its accuracy is significantly increased. The trend for the other models appear to be flatter and more linear indicating the positive correlation we observed in the original plot. We believe the bias-realism relationship suggests that future neural compression models should consider how to balance the increase of bias and loss of realism as compression bitrate decreases.

## 4.4 CAN USING A BALANCED DATASET REMOVE THE BIAS?

As highlighted in Section 4.1, the CelebA dataset is infamously racially imbalanced, potentially leading to bias in downstream tasks. This motivates the exploration of utilizing a racially balanced dataset for training neural image compression models. We utilize the FaceARG dataset and the FairFace dataset to train our models and repeat our experiments from Section 4.2. First, we highlight scenarios in which training neural compression models with the FaceARG dataset reduces bias. As presented in Figure 7(a), the *Joint* model trained on the racially balanced FaceARG dataset shows lower levels of bias in intermediate bitrates compared to the CelebA counterparts. This difference, however, is not explicit, and the trend of bias increasing with decreasing rates still exists. These slightly vary across other neural compression architectures and are presented in Appendix F. While bias is still present in this setting, these results suggest that leveraging a racially balanced training set for the neural compression model can reduce bias.

However, leveraging another racially balanced dataset, FairFace, provides alternative insight. As we observe in Figure 7(b), the FairFace dataset does not improve, and in some cases increases bias, despite also being racially balanced. We highlight that this can be due to the imbalanced of the phenotype distribution within the races themselves. This lack of phenotype variability within racial

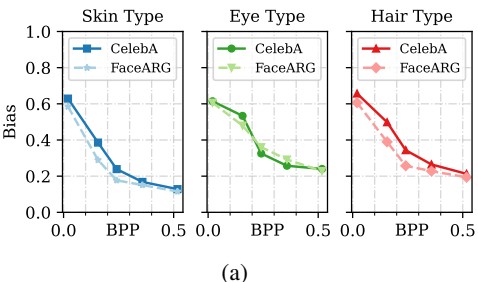 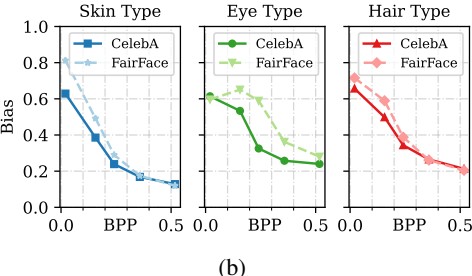

(a)                                                  (b)

Figure 7: (a) Using a racially balanced dataset (FaceARG) helps reduce the bias until extremely low bitrates less than 0.1 bpp. However, the general trend of bias increasing with decreasing bitrate is consistent across 2 datasets. (b) Using FairFace does not reduce bias and in cases increases bias.

groups can make certain phenotypes more difficult to preserve, which can lead to bias. This finding is consistent with that of Cherepanova et al. (2023), that class-balanced learning does not necessarily lead to fair classification. Additionally, the amplification of bias could be attributed to the facial orientation differences of the FairFace dataset (Laszkiewicz et al., 2024), in which images with more variable poses make reconstructions at lower rates, lower quality. We conclude that training with a balanced dataset can reduce bias in some cases but is not a sufficient bias mitigation strategy. We believe that this strongly motivates the construction of datasets that are balanced beyond race (e.g. phenotype level bias) to further reduce bias. Additionally, this motivates algorithmic methods for bias mitigation in neural image compression architectures, some of which we discuss in Section 5.

## 5 CONCLUSION AND DISCUSSION

We present a general framework to investigate the bias of neural image compression models. Using this framework we reveal bias in phenotype loss under low-rate neural compression, notably for African individuals' *skin* and *hair types* and Asian individuals' *eye types*. Additionally, we highlight bias is consistent across neural compression models. We explore the relationship between bias and realism and reveal a linear correlation within rates of one model but a trade-off across models. Finally, we demonstrate that racially balancing the dataset can help alleviate bias in certain scenarios but is not a sufficient mitigation strategy. This pioneering analysis of bias in low-rate neural image compression prompts further exploration of the domain. Future research directions include:

**Bias Mitigation** With bias present in neural compression models, a necessary future step is to explore how to mitigate this bias. As highlighted in Section 4.4, solely balancing the training data cannot fully eliminate the bias of the compression models. This motivates algorithmic methods for reducing bias in neural compression architectures. First, since neural compression can be viewed as image-to-image models with information bottlenecks, an interesting future direction is exploring how traditional fair models from the standard image-to-image space Tanjim et al. (2022) translate to the neural compression domain. Another possibility could be to adopt bias mitigation techniques designed from representation learning (Zemel et al., 2013; Louizos et al., 2015; Creager et al., 2019) to the neural compression domain, as neural compression can be viewed as a rate-constrained version of representation learning. Other methods could explore leveraging components from fairness-aware generative models (Xu et al., 2018; Friedrich et al., 2023) to design fair neural image compression models. Additionally, Tschannen et al. (2018) proposes a distribution-preserving neural compression model, which, when combined with a racially balanced training set, could yield interesting insights into constructing a fair neural compression system.

**Isolating bias** For evaluation, we utilize a single phenotype classifier across different bitrates. This allows us to isolate the bias of the classifier by examining the performance differences across difference rates. Future work can further investigate isolating the bias of the phenotype classifier by leveraging a fair classifier. Dooley et al. (2023) demonstrate that bias can be inherent to the classifier architecture and that fair architectures can be found through neural architecture search. Exploring a fair architecture for neural compression is an interesting future direction. Additionally, emerging information theoretic techniques (Goldfeld & Greenewald, 2021; Goldfeld et al., 2022; Wongso et al., 2022; 2023; Tax et al., 2017; Wibral et al., 2017; Dutta et al., 2020; Dutta & Hamman, 2023) can be explored to further decouple bias in the encoder and decoder of neural compression architectures.

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

# A DATASET DETAILS

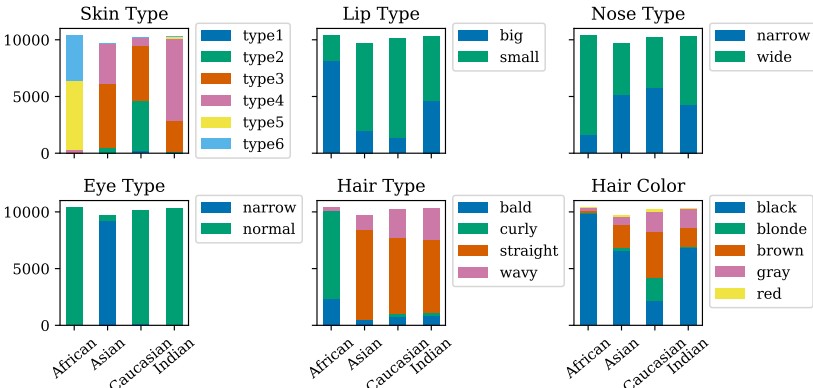

Figure A.1: Distribution of phenotype classes for each category across racial groups in RFW dataset.

We observe that across the dataset, certain phenotypes occur at different rates for different races. The distributions of *skin type*, *hair type*, and *hair color* phenotypes are dependenet on racial group. The African group has predominantly type 5 and type 6 skin, curly hair, and black hair. The Asian group has predominantly type 3 and type 4 skin, straight hair, and black hair. The Caucasian group has predominantly type 2 and type 3 skin, with straight hair and a balanced hair color distribution. The Indian group has predominantly type 3 and type 4 skin, straight hair, and black hair. Additionally, the eye type labels are extremely imbalanced within each racial group with nearly all Asian images labelled as narrow and nearly all non-Asian images labelled as wide. The *lip type* and *nose type* distributions appear relatively balanced within each racial group.

**Bias in Facial Image Datasets** Machine learning models trained on biased datasets tend to inherit and perpetuate those biases, resulting in skewed performance across different demographic groups. Many large-scale facial image databases are disproportionately biased toward individuals with lighter skin tones, underrepresenting those with darker skin (Merler et al., 2019). For instance, widely used datasets like CelebA (Liu et al., 2018), LFW (Huang et al., 2008), and UTK-Face (Zhang et al., 2017) reflect significant demographic imbalances. Beyond skin tone, other attributes such as gender and age are also prone to bias in representation. Numerous studies have explored how these biases in datasets affect the performance of downstream models, particularly in terms of fairness across demographic groups (Drozdowski et al., 2020; Buolamwini & Gebru, 2018; Hupont & Fernández, 2019). In response, recent efforts have focused on creating more diverse and discrimination-aware facial image datasets, such as FairFace (Kärkkäinen & Joo, 2019), Racial Faces in-the-Wild (RFW) (Wang et al., 2019), and FaceARG (Darabant et al., 2021), to reduce model biases and improve fairness. While these datasets reduce bias in terms of racial representation, they do not fully eliminate all forms of bias. In this paper, we focus on the facial phenotypes within the RFW dataset, which offers a relatively balanced racial composition. However, it remains imbalanced at the phenotype level, a limitation that will be explored in detail in the paper.

## B    TRADITIONAL DISTORTION METRICS

We present the PSNR, SSIM, and LPIPS distortion curves for all models trained on the CelebA dataset in Figures B.1, B.2, and B.3 respectively.

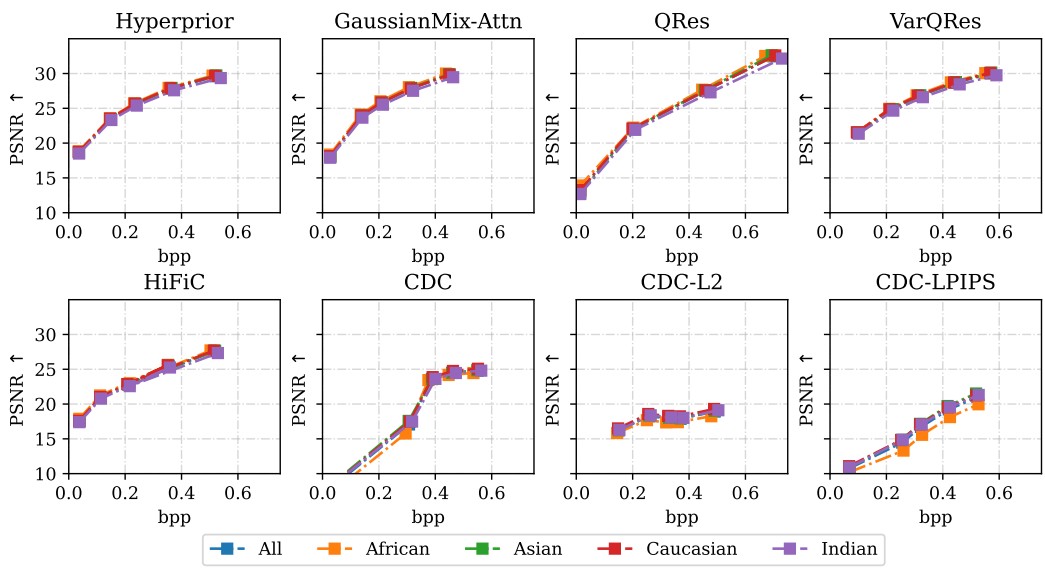

Figure B.1: PSNR rate-distortion curves for all neural compression models trained on the CelebA dataset.

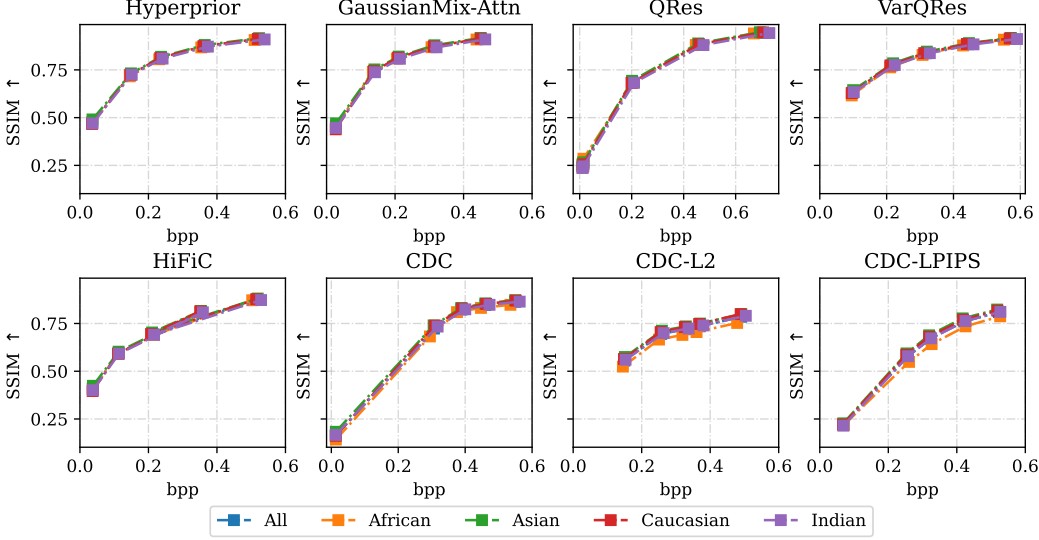

Figure B.2: SSIM rate-distortion curves for all neural compression models trained on the CelebA dataset.

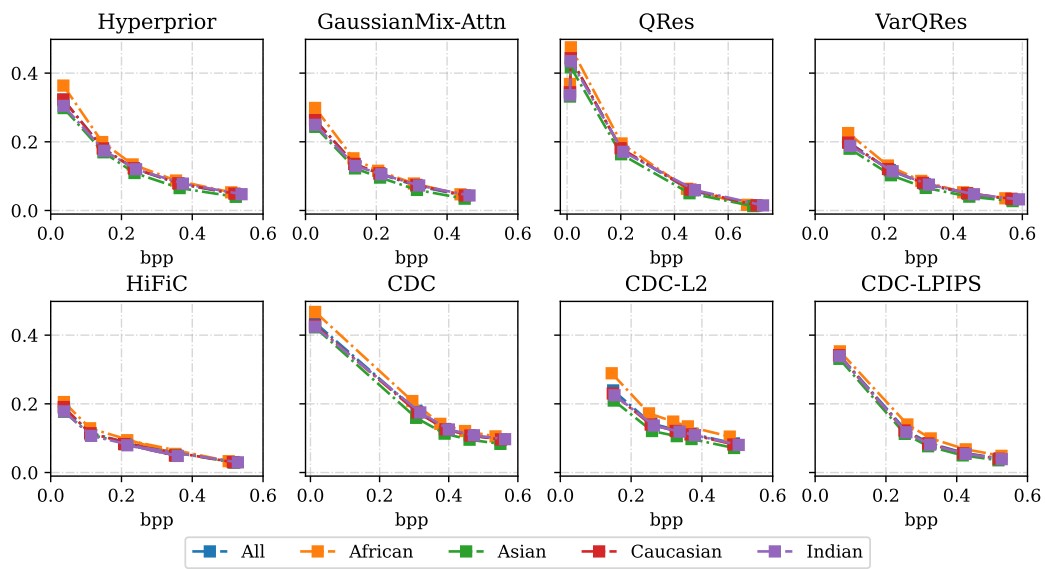

Figure B.3: LPIPS rate-distortion curves for all neural compression models trained on the CelebA dataset.

## C  TRAINING DETAILS

### C.1  PHENOTYPE CLASSIFIER

We train ResNet18 models He et al. (2016) for facial phenotype classification from scratch. The classifiers retain the ResNet18 backbone and include a classification head for classifying the specific attribute. We trained the separate phenotype classifier models for up to 50 epochs, employing early stopping with patience of 5 epochs. We use cross entropy loss and optimize the models with the stochastic gradient descent optimizer, a fixed learning rate of 0.01, and a fixed batch size of 32. To evaluate each compression model at different compression rates, we train the models on decompressed images from each of the evaluated neural compression models with different compression rates separately, using the provided dataset annotations. We report the average results over 5 runs with different random seeds for all of our experiments.

### C.2  NEURAL COMPRESSION MODELS

For models *Hyperprior*, *Joint*, and *GaussianMix-Attn*, we adopt the implementations from the CompressAI (Bégaint et al., 2020) library. For the other models, we adopt the implementation provided by the authors (Duan et al., 2023b; Mentzer et al., 2020; Yang & Mandt, 2023) or publicly available implementations [1]. For the CompressAI neural compression models, we train for 1000 epochs with an early stopping patience of 50 epochs. We use a batch size of 64 and an initial learning rate of 0.0001. For the rest of the parameters, we leave them as they are implemented in the CompressAI repository. For the *QRes* (Duan et al., 2023b), *VarQres* (Duan et al., 2023a), *HiFiC*(Mentzer et al., 2020) and *CDC*(Yang & Mandt, 2023) implementations, we follow the training procedure from the papers.

## D  RACIAL BIAS IN DEGRADATION

---

[1]https://github.com/Justin-Tan/high-fidelity-generative-compression

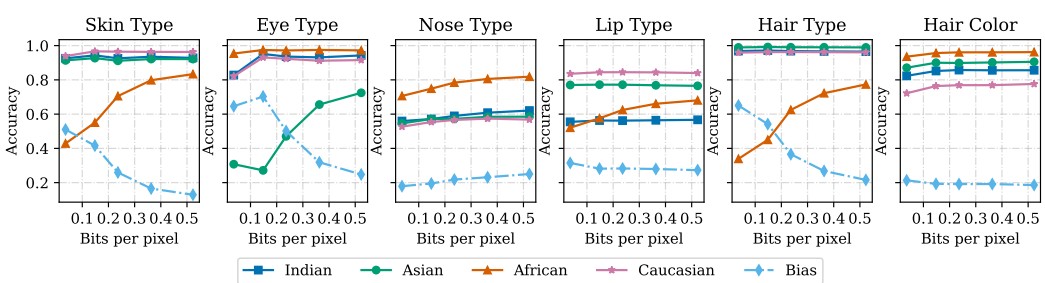

Figure D.1: Bias in phenotype degradation for the *Hyperprior* Model trained on CelebA

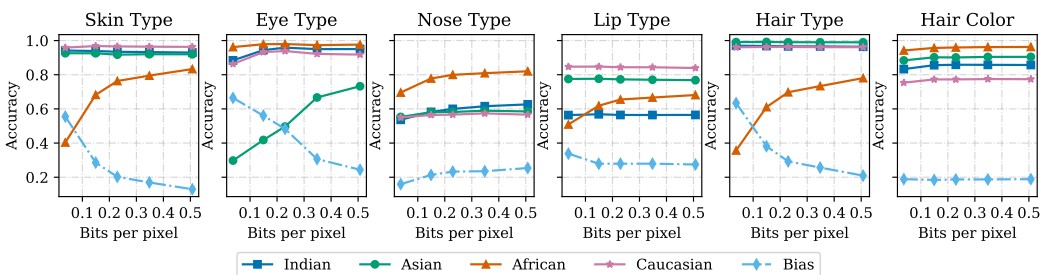

Figure D.2: Bias in phenotype degradation for the *Hyperprior* Model trained on FaceARG

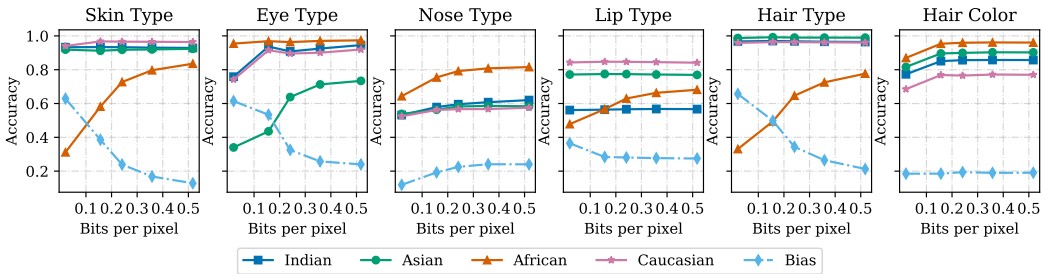

Figure D.3: Bias in phenotype degradation for the *Joint* Model trained on CelebA

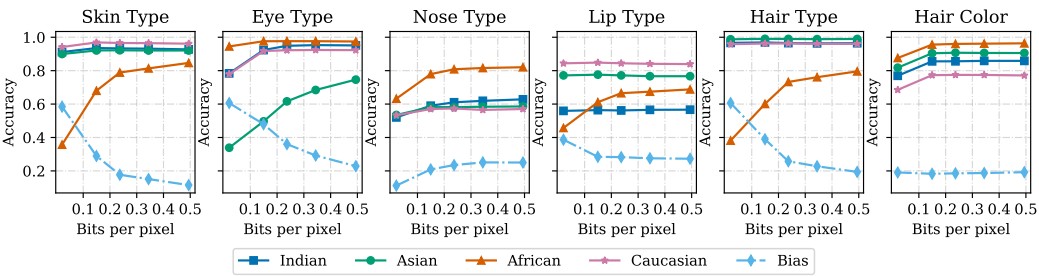

Figure D.4: Bias in phenotype degradation for the *Joint* Model trained on FaceARG

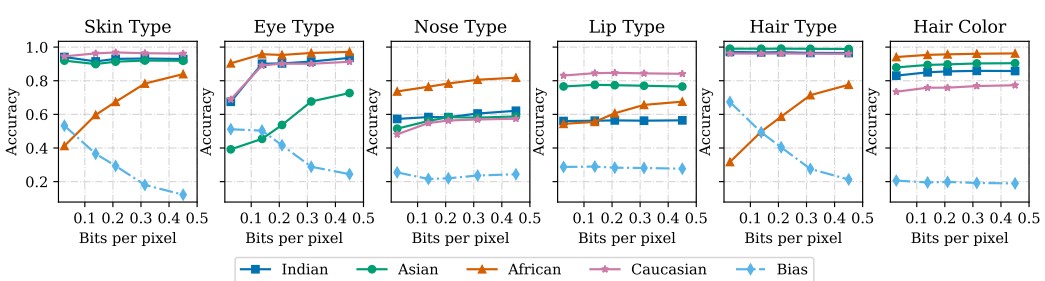

Figure D.5: Bias in phenotype degradation for the *GaussianMix-Attn* Model trained on CelebA

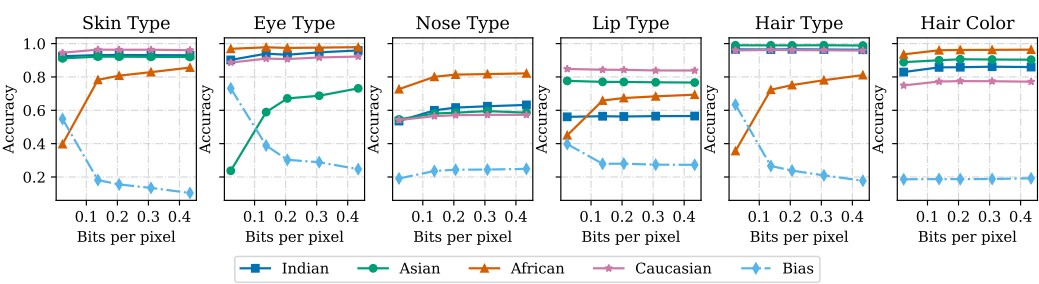

Figure D.6: Bias in phenotype degradation for the *GaussianMix-Attn* Model trained on FaceARG

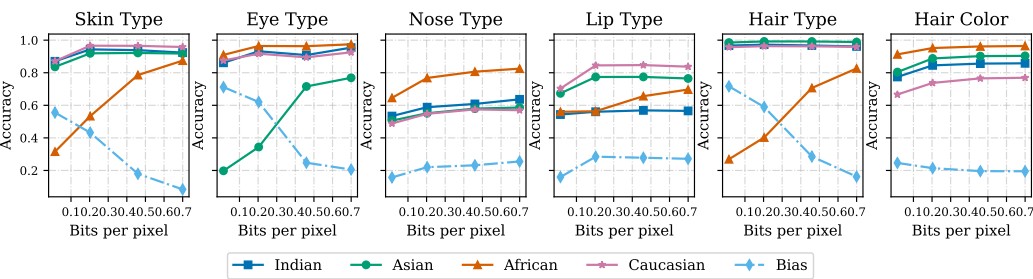

Figure D.7: Bias in phenotype degradation for the *QRes* Model trained on CelebA

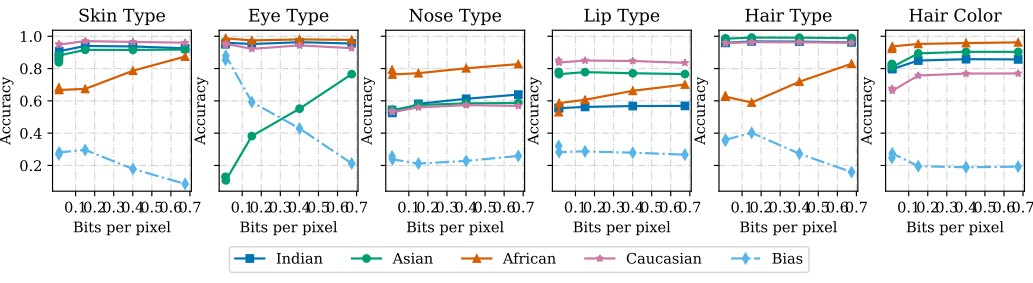

Figure D.8: Bias in phenotype degradation for the *QRes* Model trained on FaceARG

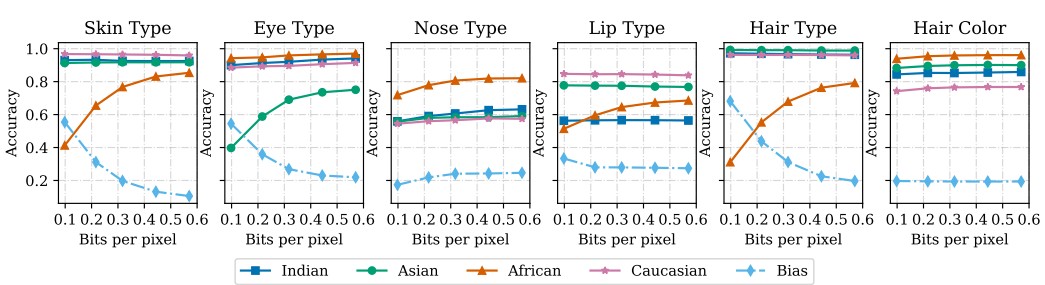

Figure D.9: Bias in phenotype degradation for the *VarQRes* Model trained on CelebA

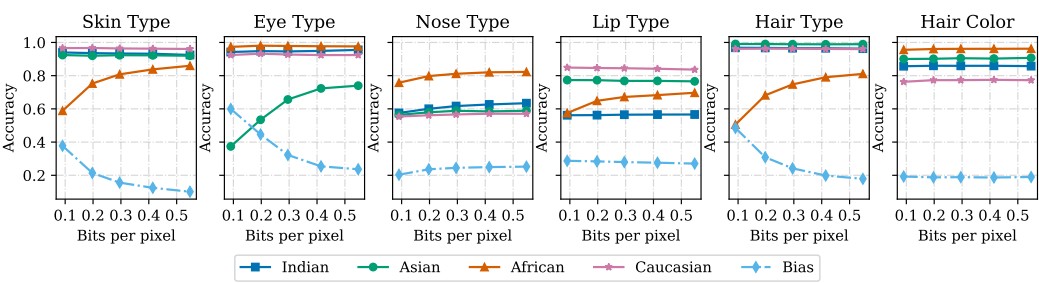

Figure D.10: Bias in phenotype degradation for the *VarQRes* Model trained on FaceARG

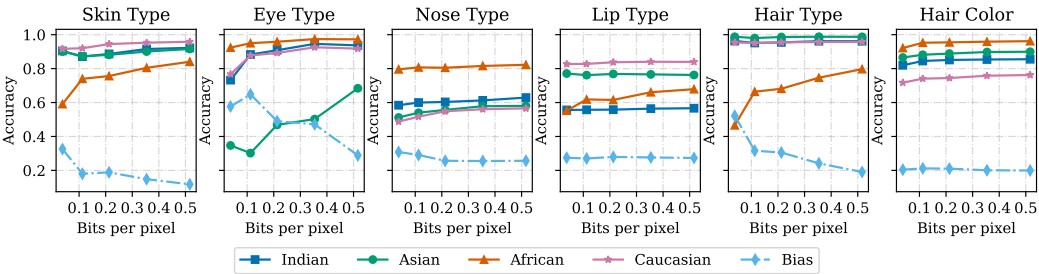

Figure D.11: Bias in phenotype degradation for the *HiFiC* Model trained on CelebA

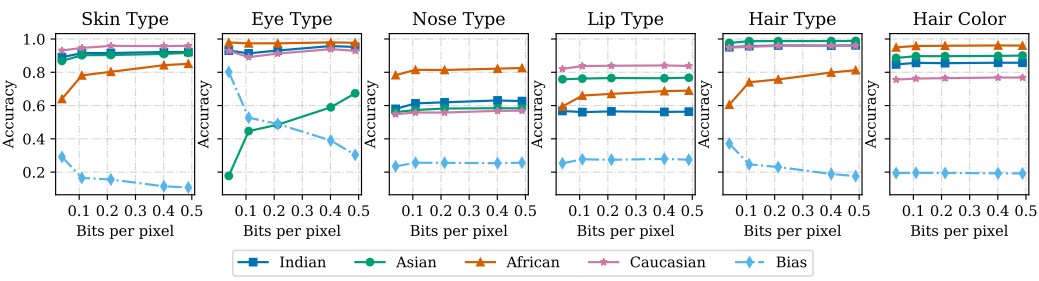

Figure D.12: Bias in phenotype degradation for the *HiFiC* Model trained on FaceARG

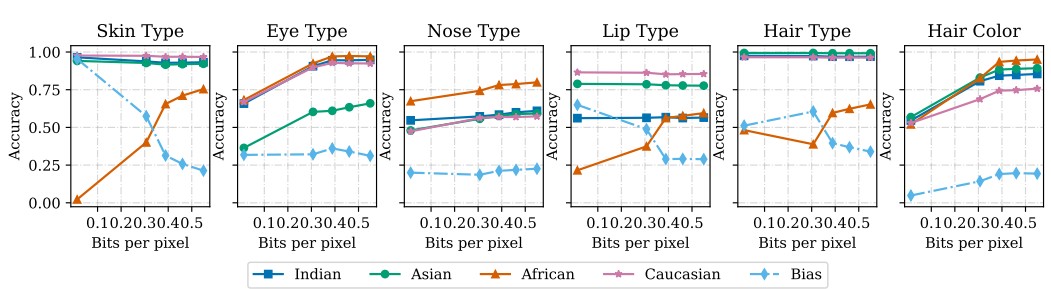

Figure D.13: Bias in phenotype degradation for the *CDC* Model trained on CelebA

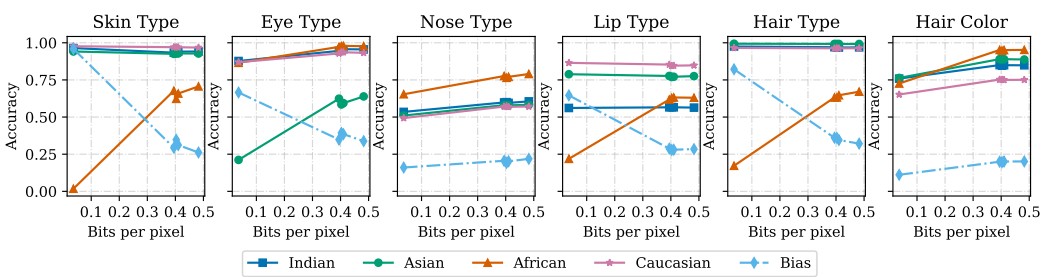

Figure D.14: Bias in phenotype degradation for the *CDC* Model trained on FaceARG

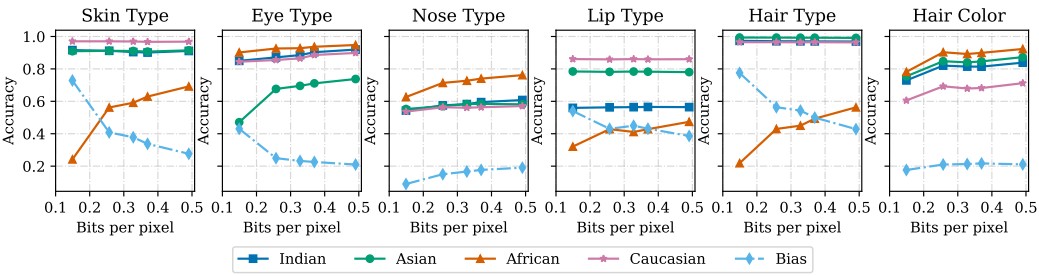

Figure D.15: Bias in phenotype degradation for the *CDC-L2* Model trained on CelebA

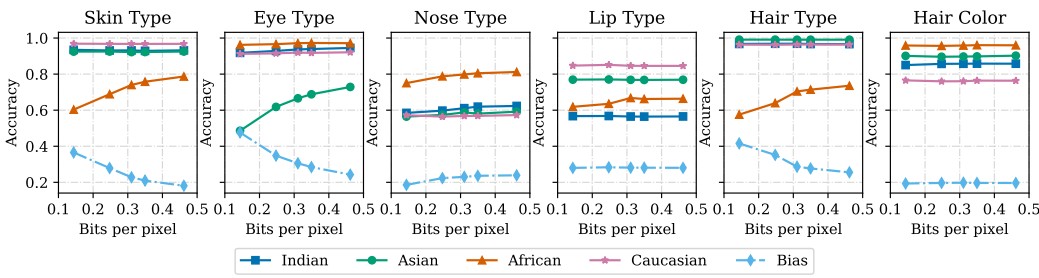

Figure D.16: Bias in phenotype degradation for the *CDC-L2* Model trained on FaceARG

Table C.1: **Neural Compression Models.** The evaluated neural compression models and variants vary in network architecture, optimization objectives, and rate control strategies.

| Model | Fixed-rate | Architecture | Realism | Rates [bpp] |
|---|---|---|---|---|
| Hyperprior | ✓ | VAE | × | 0.04 - 0.52 |
| Joint | ✓ | VAE | × | 0.02 - 0.52 |
| GaussianMix-Attn | ✓ | VAE | × | 0.03 - 0.45 |
| QRes | × | VAE | × | 0.01 - 0.70 |
| VarQRes | × | VAE | × | 0.10 - 0.51 |
| HiFiC | ✓ | VAE + GAN | ✓ | 0.04 - 0.52 |
| CDC | ✓ | Diffusion | × | 0.01 - 0.55 |
| CDC-L2 | ✓ | Diffusion | × | 0.15 - 0.50 |
| CDC-LPIPS | ✓ | Diffusion | ✓ | 0.07 - 0.52 |

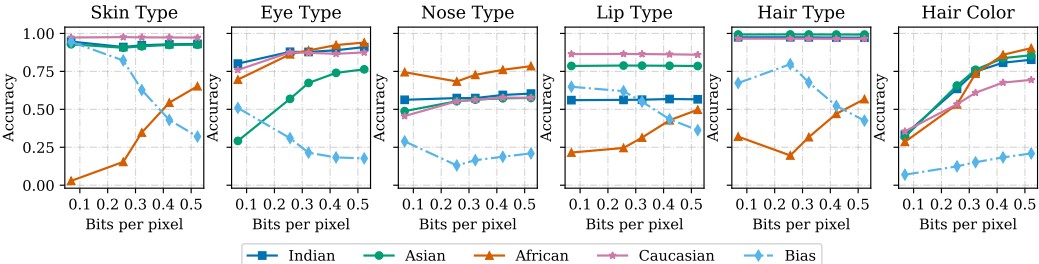

Figure D.17: Bias in phenotype degradation for the *CDC-LPIPS* Model trained on CelebA

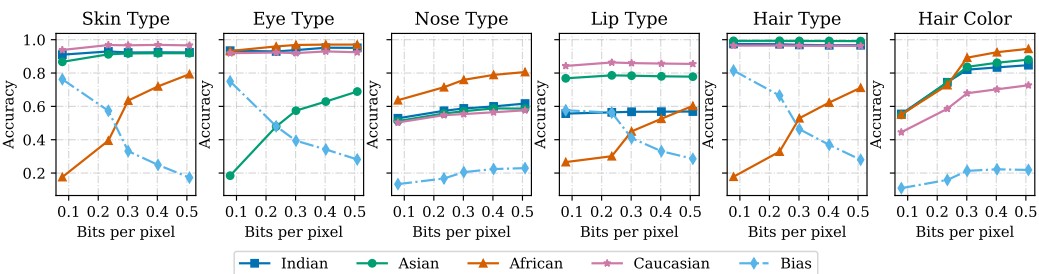

Figure D.18: Bias in phenotype degradation for the *CDC-LPIPS* Model trained on FaceARG

# E  EVALUATION OF CLASSIFIER EFFECTIVENESS

In this section, we provide further support for the effectiveness of the phenotype classifiers.

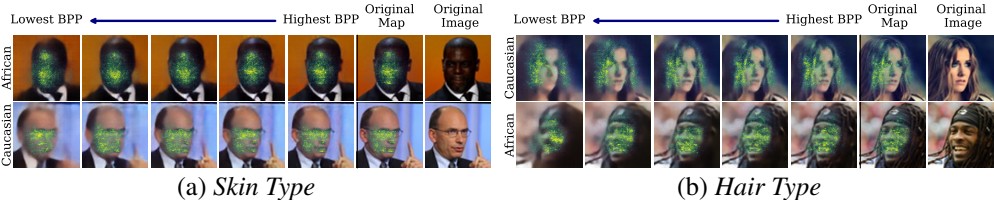

(a) *Skin Type*         (b) *Hair Type*

Figure E.1: Saliency maps at varying compression rates for African and Caucasian examples using the *VarQRes* model trained on the CelebA dataset. (a) Saliency maps for *skin type* classification. The classifier is able to recognize the general area of interest for classifying the skin type. (b) Saliency maps for *hair type* classification, where the classifier accurately locates the hair region for the Caucasian example, but fails to focus on the hair region in the African image, even in the raw image space.

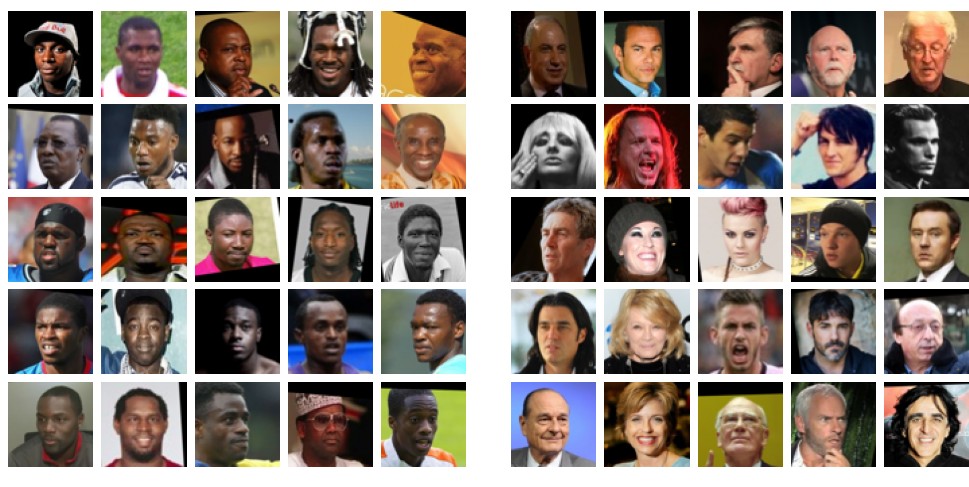

(a) African Sample                    (b) Caucasian Sample

Figure E.2: Qualitative observation of the distribution of images in the (a) African and (b) Caucasian groups. The Caucasian group exhibits a more gender-balanced set of facial images compared to the African group. Additionally, many images of African individuals feature headwear, which may complicate the classification of *hair type* within this group.

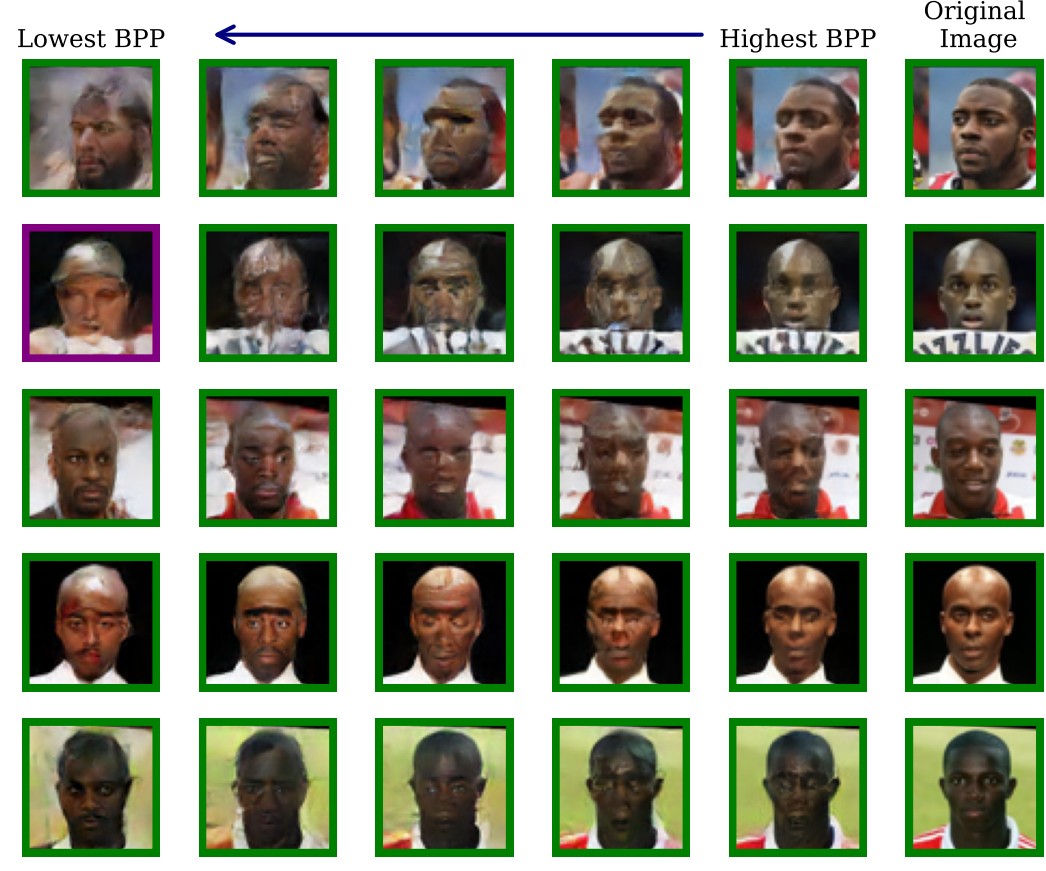

Figure E.3: *HiFiC* preserves *skin type* well. However, it introduces extra image details.

# F    TRAINING WITH A BALANCED DATASET

In Figure F we present the impact of using a balanced training set FaceARG on racial bias in phenotype degradation.

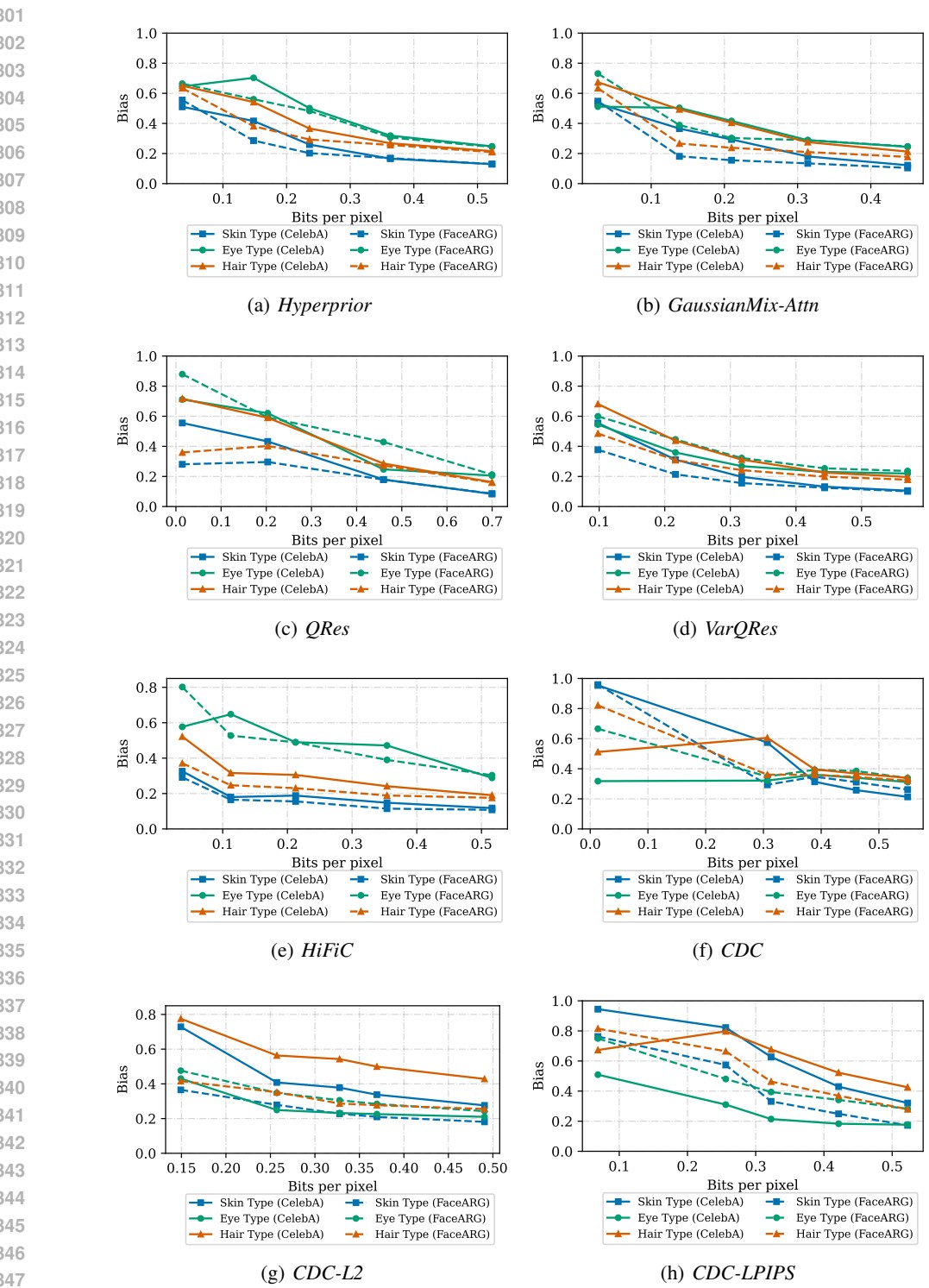

Figure F.1: Impact on phenotype degradation bias of racially balanced or imbalanced datasets

# G  BIAS-REALISM RELATIONSHIP

In Figure G and Figure G we present FID vs bias figures for all the phenotypes.

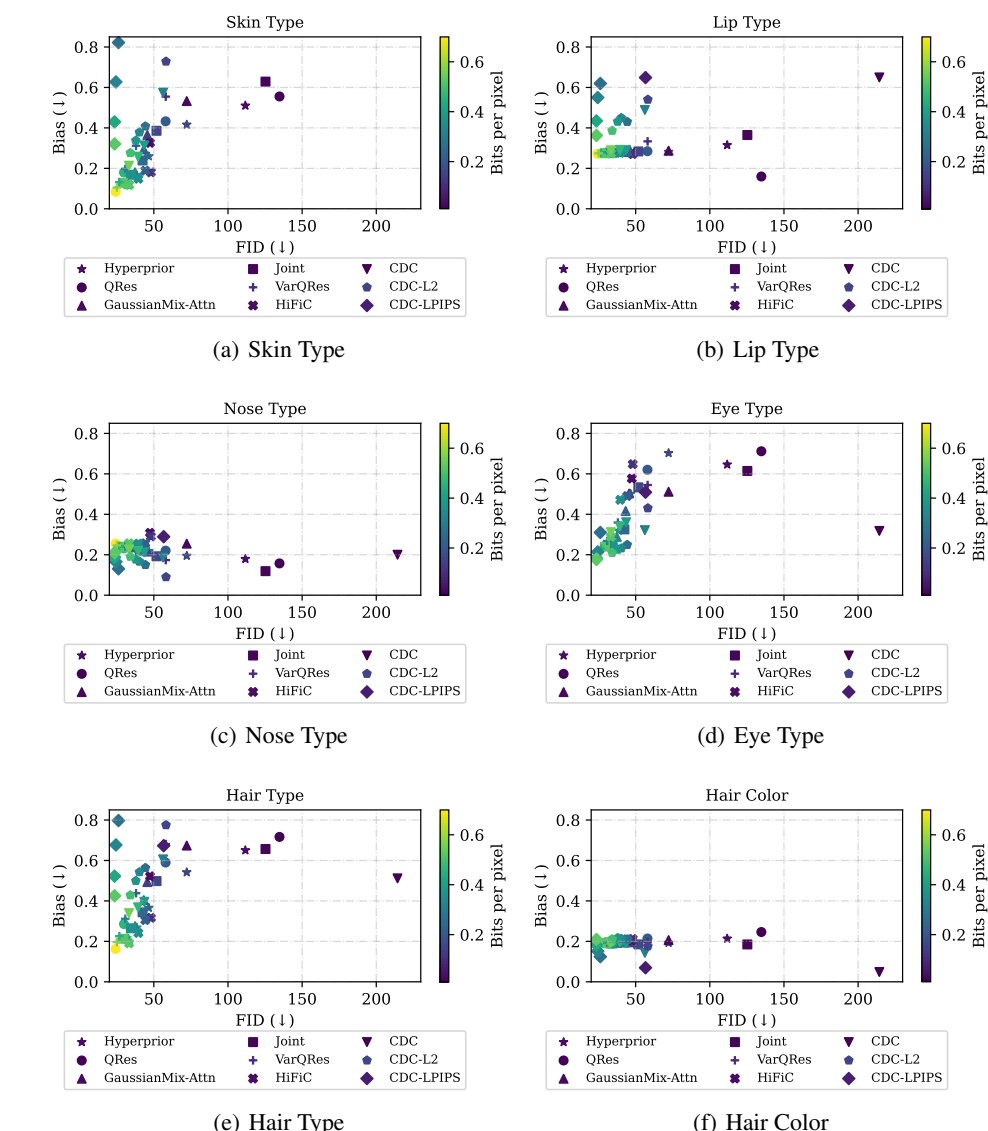

Figure G.1: Bias-realism relationship for models trained on CelebA

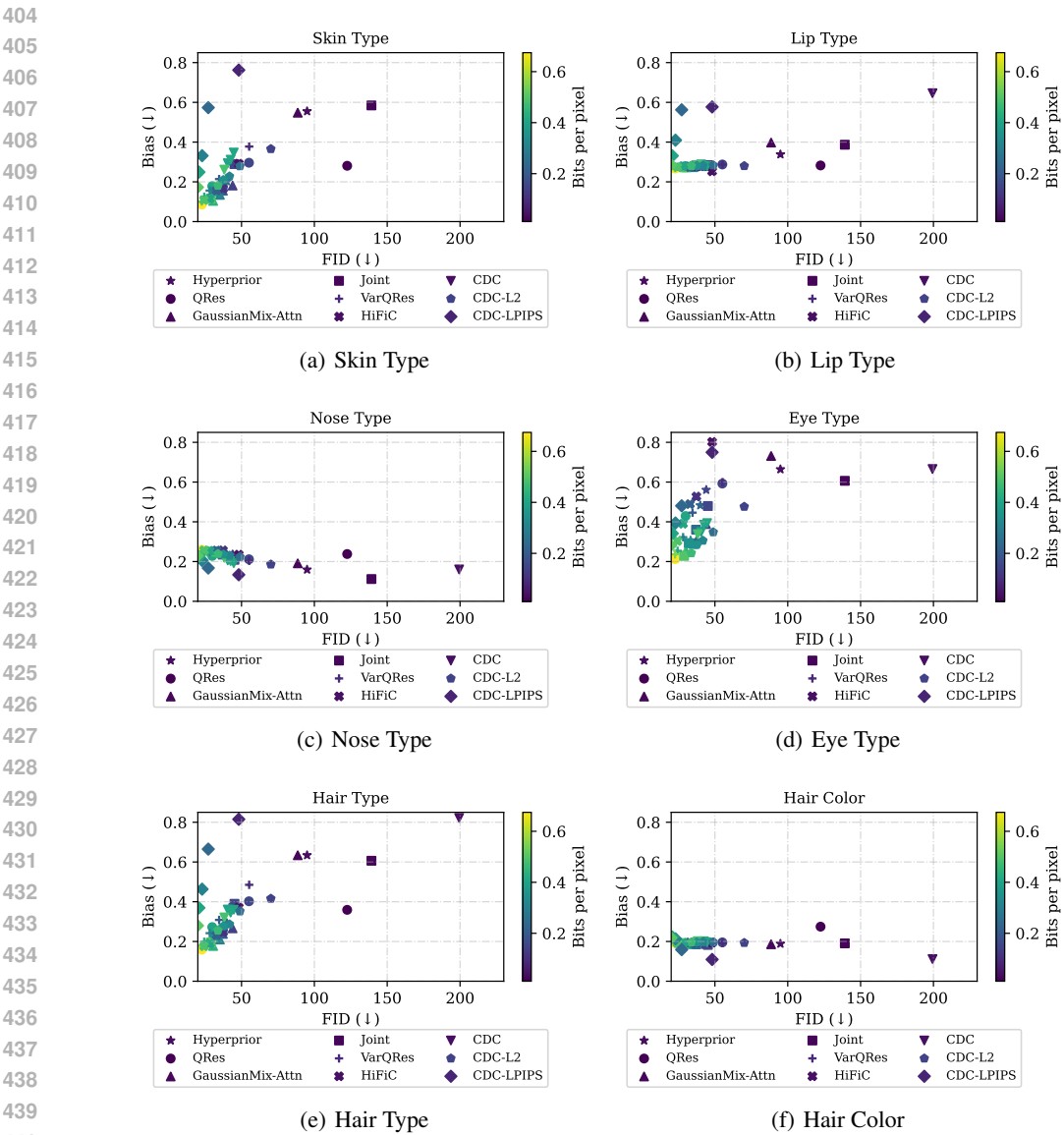

Figure G.2: Bias-realism relationship for models trained on FaceARG

# H  TRAINING WITH AFRICAN-ONLY IMAGES

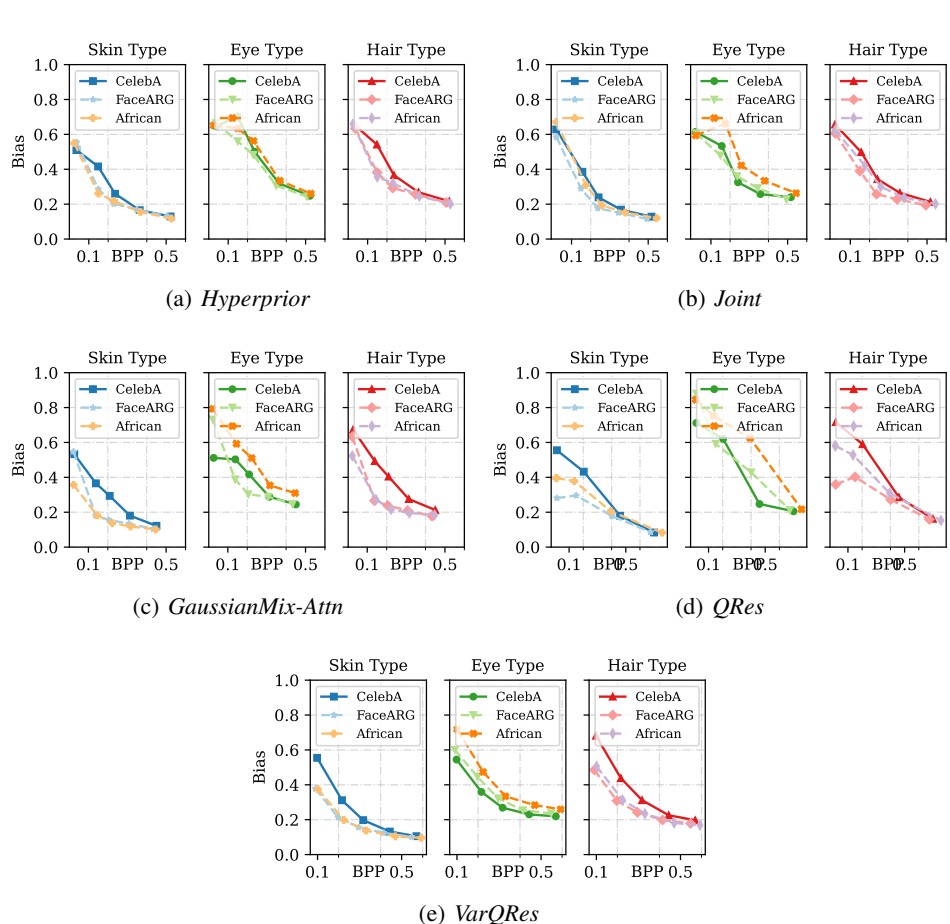

Figure H.1: Using the african-only subset from FaceARG helps reduce bias in one model (*GaussianMix-Attn*), but doesn't have significant impact on other models.

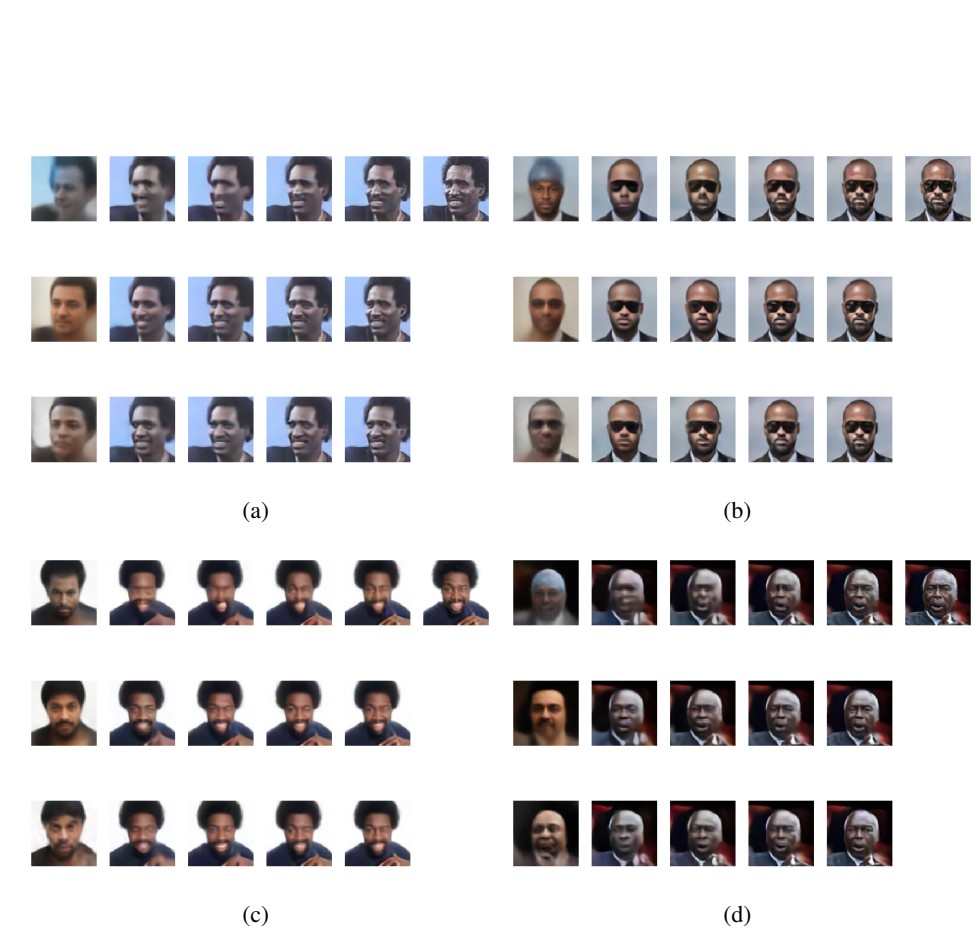

(a)                       (b)

(c)                       (d)

Figure H.2: In each subfigure, the reconstructions are from models trained with CelebA (top), FaceARG (middle), and African subset from FaceARG (bottom). The bitrate reduces from right to left, with rightmost image the original image. (a) and (b): Examples of training with african only reduces skin type bias. (c) and (d): Examples of skin type bias still exists after training with african only images.

# I  FREQUENCY DISTORTION

We are interested in understanding how each neural compression model distort different frequency components in the image. The figures below plots the percentage of reduction in signal magnitude in the frequency domain. We can observe different overall pattern across neural compression models, but the patterns across races are consistent within each models. This means the phenotype classifier is not leveraging any discrepancy in frequency distortion across races.

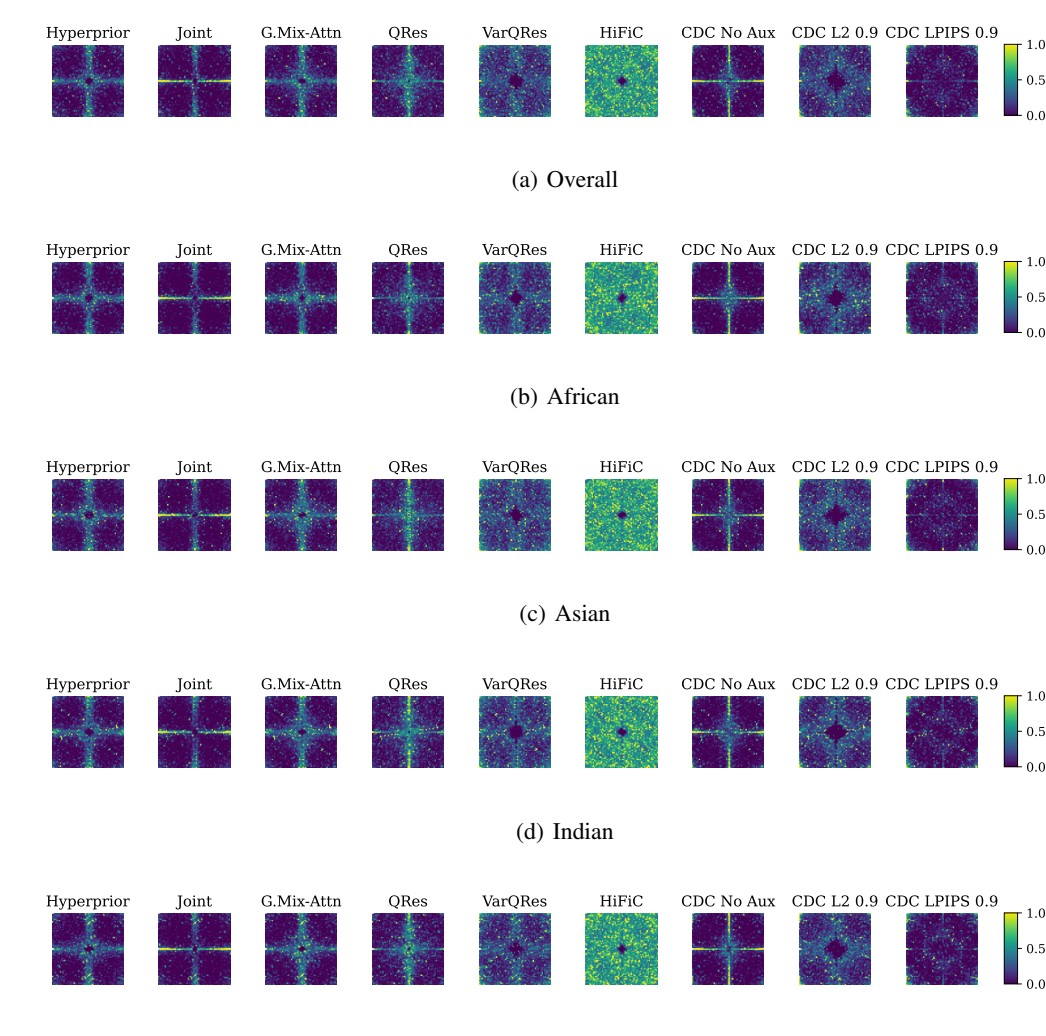

Figure I.1: Frequency degradation map for different neural compression models.

## J    RACIAL BIAS IN JPEG CODEC

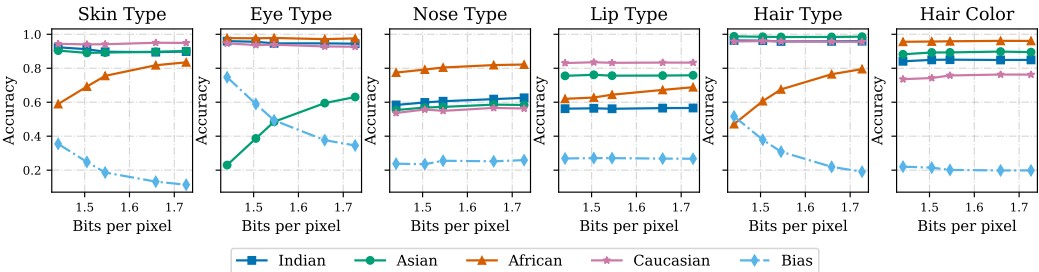

Figure J.1: Bias in phenotype degradation in JPEG

