# OpenReview forum: "Gone With the Bits: Revealing Racial Bias in Low-Rate Neural Compression for Facial Images"
_ICLR.cc/2025/Conference — ICLR 2025 Conference Withdrawn Submission_

### Official Review · Reviewer_VaCJ · 2024-10-23

**Soundness:** 3
**Presentation:** 3
**Contribution:** 2
**Rating:** 5
**Confidence:** 4

**Summary:**

This paper investigates racial bias in neural compression models for facial image compression, particularly at low bitrates. The authors demonstrate that traditional distortion metrics are insufficient for capturing racial bias, which manifests in noticeable degradation of facial features, especially for darker-skinned individuals. They examine the relationship between bias, model architecture, and visual realism, and show that while balancing the training dataset can help reduce bias, it does not fully eliminate it.

**Strengths:**

1. This paper investigate bias in neural compression models, bringing attention to an underexplored area of fairness in AI. The authors reveal clear biases, particularly in skin-type degradation.

2. The raised issue is very noteworthy and worth investigating, as it holds certain value for the generalization and reliability of neural compression methods.

**Weaknesses:**

1. One of the major concern is that: while the paper identifies the presence of bias, the essential reasons are not thoroughly explored and mitigation strategies suggested (like dataset balancing) are not shown to be completely effective.

2. The experiments demonstrate that balancing the training dataset can help but does not fully mitigate the bias.  If the dataset balance is not working well. Network architecture’s impact on bias could be critical and should be analyzed further.

3. Exploring the fundamental causes of bias is critical. For instance, what would the bias level and visualized results be like if the network is trained and tested on an Africa-only dataset?

**Questions:**

1. Bias is defined as the maximum difference in loss (Eq. 3 and Eq. 6). How to deal with the impact of extreme values on results and how well does this definition of bias reflect the overall dataset?

---

> ### Author Response · Authors · 2024-11-23
>
> We sincerely thank the reviewer for the valuable feedback.
> ### W1: Lack of bias mitigation techniques
> This paper is not intended as a solution-centered paper, but rather a benchmarking/evaluation paper to expose previously unexposed bias issues within neural compression models. Benchmarking papers have been instrumental in driving research in responsible machine learning. As AI continues to progress rapidly, it is crucial to uncover and understand unforeseen challenges that arise. For instance, at ICLR last year, several benchmarking papers (e.g., [1–3]) were accepted, highlighting emerging issues related to bias and privacy in large language models (LLMs). We believe that our benchmarking work provides valuable insights to the community and opens up  an important new research topic  on fair neural compression. By establishing a novel metric for evaluating bias in image compression and showing clear weaknesses across all existing neural compression techniques, this paper lays the groundwork for future advancements in this area (more details are provided in response to the next question).
> ### W2: Impact from network architecture is not fully explored
> We thank the reviewer for suggesting that the source of bias be attributable to the architecture. Through this paper, we have decoupled the bias that comes from the dataset and the bias that is inherent to the model (loss function, training regime, architecture). Across our experiments, we reveal  a general trend of bias that is consistent across multiple loss functions (MSE, LPIPS) and architectures (VAE variants, VAE with GAN, diffusion-based). From our initial data balancing experiments we demonstrate that bias is present across models both in settings where the training data set is racially balanced and imbalanced. To more thoroughly highlight our point, we present new experiments with African only training data (Appendix H). These experiments show that the bias that is introduced by model (loss function, training regime, architecture). We believe that this paper strongly motivates algorithmic methods for bias mitigation, which can be done in future works. While it is still unclear exactly which components of the models contribute to bias the most (a challenging problem), this is worthy of investigation and can be done in an extension of this work that explores algorithmic mitigation. Ultimately, we believe that this paper highlights an flaw with current neural image compression models and motivates a path of algorithmic mitigation of bias in these models.
> ### W3: Fundamental cause of bias & training with African-only datasets
> We thank the reviewer for the suggestion of training with African-only images. We have conducted experiments training neural compression models with African subset from the FaceARG dataset which has provided us with additional valuable insights. We have updated the paper with Appendix H. As the figures have shown, training with African-only images help reduce the bias in skin type in one model (GaussianMix-Attn), but not in other models. This result indicates that using an African-only dataset doesn’t completely remove the bias, and that the bias that is introduced by model (loss function, training regime, architecture), which tend to generate lighter-colored images. \
> We have also attached visualized results (Appendix Fig. H.2) to show examples of where training with African-only images helps or does not help reduce the bias in skin type. \
> This additional experiment really gave more insights to this problem, so we sincerely thank the reviewer for the suggestion!
> ### References
> [1] Gupta, Shashank, et al. "Bias Runs Deep: Implicit Reasoning Biases in Persona-Assigned LLMs." *The Twelfth International Conference on Learning Representations*.\
> [2] Belém, Catarina G., et al. "Are Models Biased on Text without Gender-related Language?." *The Twelfth International Conference on Learning Representations*.\
> [3] Staab, Robin, et al. "Beyond Memorization: Violating Privacy via Inference with Large Language Models." *The Twelfth International Conference on Learning Representations*.

---

> > ### Comment · Reviewer_VaCJ · 2024-11-25
> > **Response to the rebuttal**
> >
> > Regarding the bias phenomenon in LICs, I believe it is crucial to focus on understanding its underlying causes and identifying effective solutions. However, as pointed out by other reviewers, the paper does not sufficiently address these aspects. I hope the authors can investigate this issue more thoroughly in future work. I will maintain my score.

---

### Official Review · Reviewer_cyKR · 2024-10-28

**Soundness:** 3
**Presentation:** 3
**Contribution:** 3
**Rating:** 6
**Confidence:** 4

**Summary:**

The paper presents an analysis of racial bias in different neural compression algorithms. To measure this, the method uses face/phenotype classifiers and measure how much the classification decision is affected by the neural compression algorithm. This is similar to a rate-distortion measure where distortion is classification error instead of pixel error. The paper shows a clear bias in neural compression algorithms when trained on imbalanced datasets which is only partially mitigated by using balanced data.

**Strengths:**

The work here is interesting and timely. As neural compression continues to approach a usable state, but has yet to be deployed in any meaningful way, it is extremely important to start considering any potential bias in trained models or inherent to the algorithms. This way, the community can develop mitigations and ensure that those mitigations are used in future products. Additionally, I think the metrics used by the method to quantify bias are sensible and the finding that traditional distortion metrics, such as PSNR, do not accurately capture bias is a good result. This also makes sense since many compression techniques use pixel-error metrics as their objective.

Although there is some overlap with prior work, as discussed in the paper, I think there is significant value in extending the analysis to neural compression.

**Weaknesses:**

While the classification error metrics presented in the paper is a good start it may need some additional development to make it fully capture bias. As the authors point out: the metric is only as good as the classifier itself. If the classifier is not able to make reliable decisions then the metric could miss bias or overly assign bias. I think this topic deserves more attention. Additionally the sensitivity of classifiers to different frequency degradations (which are common for compression); this may also explain different classification results at low bitrates.

Finally, there is an entire class of compression algorithms based on Implicit Neural Representations (see SIREN [1] for one example) which train a neural compression model unique to each example. This kind of technique could help mitigate any bias but these were not tested in the paper.

1. https://arxiv.org/abs/2006.09661

**Questions:**

* How can we show better reliability of the classification metric?
* Could INR methods overcome potential bias in neural compression?

---

> ### Author Response · Authors · 2024-11-23
>
> We sincerely thank the reviewer for the valuable feedback.
> ### W1: Phenotype Classifier Evaluation
> We have additionally run human studies to analyze how humans classify skin type in facial images. We asked users to label both uncompressed original images and distorted images decoded from the lowest bitrate from the GaussianMix-Attn model. From the human studies, we observe human’s ability to discern the original skin type for african racial group drops 32%, much higher than the reduction in other races, which are at most 14%. This reflects that humans perceive distorted african facial images as lighter, which is also consistent with both our initial observation that African racial group suffer significantly from losing skin type information, and the classifier accuracy results. \
> Besides human study, we generated saliency maps to verify the validity of phenotype classifiers.(line 371). The saliency maps confirm that the classifiers are looking at relevant images features for classification.
> ### W2: Sensitivity of Classifiers to Frequency Degradations
> We studied the frequency domain degradation introduced by different neural compression models and edited the paper to include results in Appendix I. The figures plots the percentage of reduction in signal magnitude in the frequency domain. We can observe different overall patterns across neural compression models, but the patterns across races are consistent within each model. This suggests that the phenotype classifier is not leveraging any discrepancy in frequency distortion across races.
> ### W3: Implicit Neural Representations
> We thank the reviewer for mentioning INR as a potential neural compression method with no bias. INR models overfit a sample to a small network and transmit the network parameters as the bitstream. Even though it does not suffer from imbalanced dataset distribution, the existence of bias in INR models are not fully studied and is worth further exploration as a potential mitigation strategy. Due to the time limit, we are not able to do a full evaluation of INR models, but this is definitely an exciting future direction!

---

### Official Review · Reviewer_MyYt · 2024-11-02

**Soundness:** 2
**Presentation:** 3
**Contribution:** 2
**Rating:** 3
**Confidence:** 4

**Summary:**

This paper investigated the racial bias problems existing in learned image compression. The authors built a framework to systematically examine the extent to which racial bias occurs in compression. Based on the evaluating framework, they proposed a classification-accuracy-based loss function to better reveal the bias. The correlation between bias, model architecture and image realism has been measured. They also show that utilizing a racially balanced training set cannot fix the problem.

**Strengths:**

1.	The paper has a clear problem definition, constructed a reasonable evaluation framework.
2.	Existing experiments have proved the existence of the problem from multiple angles to a certain extent

**Weaknesses:**

1.    The paper seemly shows few contributions to the compression community. Since the paper just proposes the racial bias problem existing in learned image compression but provides no solution from the compression perspective. Similar evaluation schemes seem to be applicable to any field. Have you considered proposing compression-specific bias mitigation techniques?
2.    It seems that the bias problem is mainly attributed to the dataset and optimizing method. But the authors only focus on the data-related reasons and do not explore the impact of model optimization methods on this issue. It seems unconvincing to simply attribute the difference in model bias to the difference in model architecture. Have you considered analyzing how different loss functions or training regimes impact bias in compression models?
3.    The authors did not provide bias analysis results for images decoded by traditional codecs like JPEG, HM and VTM. The optimization of traditional codecs is not affected by the distribution of the dataset and should not lead to bias. If this experiment can be provided, it will promote our understanding of this problem. Please estimate traditional codec results at equivalent bitrates using the same bias evaluation framework.
4.    The author used the accuracy of the classification model to evaluate the loss of image attributes at low bitrates. However, the classification model was learned on undistorted images, and whether it can accurately classify features on distorted images is unverified. Additional experiments should be conducted in this regard to enhance the persuasiveness of the bias-related conclusion. For example, you could compare classifier’s results with human evaluations on a subset of distorted images and report the accuracy.

**Questions:**

Please refer to above weakness part.

---

> ### Author Response · Authors · 2024-11-23
> **Reply to Reviewer MyYt - Part 1**
>
> We sincerely thank the reviewer for the valuable feedback.
> ### W1: About lack of bias mitigation techniques
> This paper is not intended as a solution-centered paper, but rather a benchmarking/evaluation paper to expose previously unexposed bias issues within neural compression models. Benchmarking papers have been instrumental in driving research in responsible machine learning. As AI continues to progress rapidly, it is crucial to uncover and understand unforeseen challenges that arise. For instance, at ICLR last year, several benchmarking papers (e.g., [1–3]) were accepted, highlighting emerging issues related to bias and privacy in large language models (LLMs). We believe that our benchmarking work provides valuable insights to the community and opens up  an important new research topic  on fair neural compression. By establishing a novel metric for evaluating bias in image compression and showing clear weaknesses across all existing neural compression techniques, this paper lays the groundwork for future advancements in this area (more details are provided in response to the next question).
> ### W2: Impact of loss functions and training regimes
> We thank the reviewer for suggesting that the source of bias be attributable to loss functions or training regimes. Through this paper, we have decoupled the bias that comes from the dataset and the bias that is inherent to the model (loss function, training regime, architecture). Across our experiments, we reveal  a general trend of bias that is consistent across multiple loss functions (MSE, LPIPS) and architectures (VAE variants, VAE with GAN, diffusion-based). From our initial data balancing experiments we demonstrate that bias is present across models both in settings where the training data set is racially balanced and imbalanced. To more thoroughly highlight our point, we present new experiments with African only training data (Appendix H). These experiments show that the bias that is introduced by model (loss function, training regime, architecture). We believe that this paper strongly motivates algorithmic methods for bias mitigation, which can be done in future works. While it is still unclear exactly which components of the models contribute to bias the most (a challenging problem), this is worthy of investigation and can be done in an extension of this work that explores algorithmic mitigation. Ultimately, we believe that this paper highlights an flaw with current neural image compression models and motivates a path of algorithmic mitigation of bias in these models.
> ### W3: Comparison against traditional codecs
> We thank the reviewer for mentioning traditional codecs. In this paper, we focus on low bitrate regimes that can be as low as 0.1bpp. This is a bitrate that JPEG cannot operate in. In order to compare to JPEG, we have compressed images using the lowest quality levels (implemented with Pillow). The bitrates achieved by JPEG are greater than 1 bpp, which is significantly greater than the bitrates we evaluate neural compression models in. We conducted the same bias analysis experiments from Section 4.2 for the JPEG codec. We have updated paper with Appendix J. From the figure, we can see that JPEG codec experiences similar bias towards the African racial group. This finding resonates with the existing literature [4], in which the authors conclude that facial images of individuals with darker skin tones suffer from higher error rates in facial recognition tasks after JPEG compression. These results reenforce the presence of bias in JPEG compression settings as well as further justify the use of our phenotype classifier as an evaluation metric.
> ### References
> [1] Gupta, Shashank, et al. "Bias Runs Deep: Implicit Reasoning Biases in Persona-Assigned LLMs." *The Twelfth International Conference on Learning Representations*.\
> [2] Belém, Catarina G., et al. "Are Models Biased on Text without Gender-related Language?." *The Twelfth International Conference on Learning Representations*.\
> [3] Staab, Robin, et al. "Beyond Memorization: Violating Privacy via Inference with Large Language Models." *The Twelfth International Conference on Learning Representations*.\
> [4] Yucer, Seyma, et al. "Does lossy image compression affect racial bias within face recognition?." 2022 IEEE International Joint Conference on Biometrics (IJCB).

---

> > ### Author Response · Authors · 2024-11-23
> > **Reply to Reviewer MyYt - Part 2**
> >
> > ### W4: Further Justification of the Phenotype Classifier with human evaluations
> > We have additionally run human studies to analyze how humans classify skin type in facial images. We asked users to label both uncompressed original images and distorted images decoded from the lowest bitrate from the GaussianMix-Attn model. From the human studies, we observe human’s ability to discern the original skin type for african racial group drops 32%, much higher than the reduction in other races, which are at most 14%. This reflects that humans perceive distorted african facial images as lighter, which is also consistent with both our initial observation that African racial group suffer significantly from losing skin type information, and the classifier accuracy results.

---

### Official Review · Reviewer_FHQM · 2024-11-11

**Soundness:** 3
**Presentation:** 3
**Contribution:** 2
**Rating:** 3
**Confidence:** 4

**Summary:**

This paper introduces a framework for assessing bias in neural image compression models, analyzing seven popular models and finding prevalent racial bias, manifested as unequal degradation of facial features. The study indicates that while using a racially balanced dataset helps mitigate bias, it is not a complete solution. The study indicates that while using a racially balanced dataset helps mitigate bias, it is not a complete solution.

**Strengths:**

Strength: (1)The topic of this paper is novel, and the racial bias of image compression on face data sets is studied (2)The authors have conducted quite sufficient experiments around this argument to verify that this problem does exist

**Weaknesses:**

Weakness: (1)Although the author presents a novel topic, it seems that the author did not fully explore the way to solve the problem. Using a more balanced dataset seems to be one solution, but after discussion by the authors, this approach does not completely eliminate racial bias. So, how to better solve this problem? The author needs to give further elaboration. In fact, this is the point I am most concerned about. (2)The authors used traditional metrics such as PSNR and SSIM in their experiments to reflect racial bias. However, these metrics differ significantly from human visual experience. I wonder if the authors explored more perceptual metrics, such as LPIPS or FID?

**Questions:**

Weakness: (1)Although the author presents a novel topic, it seems that the author did not fully explore the way to solve the problem. Using a more balanced dataset seems to be one solution, but after discussion by the authors, this approach does not completely eliminate racial bias. So, how to better solve this problem? The author needs to give further elaboration. In fact, this is the point I am most concerned about. (2)The authors used traditional metrics such as PSNR and SSIM in their experiments to reflect racial bias. However, these metrics differ significantly from human visual experience. I wonder if the authors explored more perceptual metrics, such as LPIPS or FID?

---

> ### Author Response · Authors · 2024-11-23
>
> We sincerely thank the reviewer for the valuable feedback.
> ### W1: Lack of bias mitigation techniques
> This paper is not intended as a solution-centered paper, but rather a benchmarking/evaluation paper to expose previously unexposed bias issues within neural compression models. Benchmarking papers have been instrumental in driving research in responsible machine learning. As AI continues to progress rapidly, it is crucial to uncover and understand unforeseen challenges that arise. For instance, at ICLR last year, several benchmarking papers (e.g., [1–3]) were accepted, highlighting emerging issues related to bias and privacy in large language models (LLMs). We believe that our benchmarking work provides valuable insights to the community and opens up  an important new research topic  on fair neural compression. By establishing a novel metric for evaluating bias in image compression and showing clear weaknesses across all existing neural compression techniques, this paper lays the groundwork for future advancements in this area.
> ### W2: Perceptual Metrics
> Thank you for mentioning perceptual metrics. We have added the LPIPS metric to capture the bias in neural compression.
> As shown in Figure 2, while LPIPS aligns more closely with human perception—indicated by the higher curve for African images compared to other, however it does not capture the difference in phenotype degradation across races. This indicates that LPIPS, like MSE and PSNR, is not a sufficient metric to capture bias in these settings. To address this limitation, we introduce the phenotype classifier task, specifically designed to detect and quantify bias that these traditional metrics overlook.
> ### References:
> [1] Gupta, Shashank, et al. "Bias Runs Deep: Implicit Reasoning Biases in Persona-Assigned LLMs." *The Twelfth International Conference on Learning Representations*.\
> [2] Belém, Catarina G., et al. "Are Models Biased on Text without Gender-related Language?." *The Twelfth International Conference on Learning Representations*.\
> [3] Staab, Robin, et al. "Beyond Memorization: Violating Privacy via Inference with Large Language Models." *The Twelfth International Conference on Learning Representations*.

---

### Author Response · Authors · 2024-12-02
**Comments to all reviewers**

We thank all the reviewers for providing valuable feedback on our paper. We appreciate the reviewers for recognizing the contributions of our paper, and for agreeing that we are revealing an important problem that should be considered in designing neural compression algorithms. Specifically, the reviewers describe our paper that reveals racial bias in neural compression as novel (FHQM), timely, and consider the raised issues very noteworthy and worth investigating (cyKR; VaCJ). The reviewers also recognize the extensive experiments (FHQM), and agree that the proposed evaluation framework is effective at capturing the bias (FHQM, MyYt).\
There are 3 common concerns that the reviewers raised, and we hope we have addressed these concerns in the comments. We’d appreciate any feedbacks to further strengthen our paper.
- The paper lacks bias mitigation approaches (FHQM W1, MyYt W1, VaCJ W1)

This paper is not intended as a solution-centered paper, but rather a benchmarking/evaluation paper to expose previously unexposed bias issues within neural compression models. By establishing a novel metric for evaluating bias in image compression and showing clear weaknesses across all existing neural compression techniques, this paper lays the groundwork for future advancements in this area.
- The paper lacks analysis on fundamental cause of bias (MyYt W2, VaCJ W3)

Through this paper, we have decoupled the bias that comes from the dataset and the bias that is inherent to the model (loss function, training regime, architecture). Across our experiments, we reveal a general trend of bias that is consistent across multiple loss functions (MSE, LPIPS) and architectures (VAE variants, VAE with GAN, diffusion-based). From our initial data balancing experiments we demonstrate that bias is present across models both in settings where the training data set is racially balanced and imbalanced. To more thoroughly highlight our point, we present new experiments with African only training data (Appendix H). These experiments show that the bias that is introduced by model (loss function, training regime, architecture).

By highlighting bias across all models, we demonstrate the presence of bias in all neural image compression settings. These results suggest that there is no single component of neural compression that we can attribute the bias to and that this bias cannot be further isolated. As there are no simple ways to isolate and easily remove bias, we believe that this paper strongly motivates algorithmic methods for bias mitigation, an entire new direction of research, which can be done in future follow-up works. Ultimately, we believe that this paper highlights an flaw with current neural image compression models and motivates a path of algorithmic mitigation of bias in these models.
- Justification of phenotype classifier-based bias analysis (MyYt W4, cyKR W1)

We have additionally run human studies to analyze how humans classify skin type in facial images. We asked users to label both uncompressed original images and distorted images decoded from the lowest bitrate from the GaussianMix-Attn model. From the human studies, we observe human’s ability to discern the original skin type for african racial group drops 32%, much higher than the reduction in other races, which are at most 14%. This reflects that humans perceive distorted african facial images as lighter, which is also consistent with both our initial observation that African racial group suffer significantly from losing skin type information, and the classifier accuracy results. Besides human study, we generated saliency maps to verify the validity of phenotype classifiers.(line 371). The saliency maps confirm that the classifiers are looking at relevant images features for classification.

We hope our additional experiments and responses address the concerns from the reviewers. We kindly request reviewers to reply whether our responses answer their questions, and would be happy to participate in further discussions. Thank you!

---

### Note · Authors · 2025-01-17

I have read and agree with the venue's withdrawal policy on behalf of myself and my co-authors.